# Evidence of reactivation of a hydrothermal system from seismic anisotropy changes

Maria Saade[1]*, Kohtaro Araragi[2], Jean Paul Montagner [1], Edouard Kaminski [1], Philippe Roux[3], Yosuke Aoki [2] & Florent Brenguier[3]

Seismic velocity measurements have revealed that the Tohoku-Oki earthquake affected velocity structures of volcanic zones far from the epicenter. Using a seismological method based on ambient seismic noise interferometry, we monitored the anisotropy in the Mount Fuji area during the year 2011, in which the Tohoku-Oki earthquake occurred ($M_w = 9.0$). Here we show that even at 400 km from the epicenter, temporal variations of seismic anisotropy were observed. These variations can be explained by changes in the alignment of cracks or fluid inclusions beneath the volcanic area due to stress perturbations and the propagation of a hydrothermal fluid surge beneath the Hakone hydrothermal volcanic area. Our results demonstrate how a better understanding of the origin of anisotropy and its temporal changes beneath volcanoes and in the crust can provide insight into active processes, and can be used as part of a suite of volcanic monitoring and forecasting tools.

[1] Institut de Physique du Globe de Paris, Sorbonne Paris Cité, Université Paris Diderot, CNRS, Paris, France. [2] Earthquake Research Institute, University of Tokyo, Tokyo, Japan. [3] Institut des Sciences de la Terre, Grenoble, France. *email: maria.saade90@gmail.com

Recent studies have shown that volcanic areas are very sensitive to stress perturbations, which can increase permeability and fluid mobility[1,2], even if distant from an earthquake epicenter[3,4]. For example, it was demonstrated that volcanic areas in Japan located 150 to 200 km from the rupture area associated with the 2011 Tohoku-Oki earthquake experienced subsidence[5]. Geodetic observations have confirmed that volcanic areas can undergo subsidence and can be affected by crustal deformation induced by large earthquakes centered even hundreds of kilometers away[5,6]. As well as crustal deformation, crustal seismic velocity reductions were observed below volcanic regions[7], and in particular for the Mount Fuji area (400 km from the epicenter). Finally, temporal changes in seismic anisotropy have also been observed in other volcanic areas worldwide[8–10], and these have been interpreted as stress changes due to magma intrusion at shallow depths. Indeed, a co-seismic change in the stress field can induce changes in the average orientation of cracks, which will affect the polarization of surface waves.[11,12]

Going one step further, we measure here the time evolution of stress-induced anisotropy in the area of Mount Fuji using noise-based seismic interferometry, for the time period covering the 2011 Tohoku-Oki earthquake. Our study also aims to identify the transfer and type of fluid that might reactivate the hydrothermal or magmatic system after the mainshock.

## Results

**Anisotropy measurements**. We processed continuous data from 18 seismic stations (Fig. 1) that were recorded in 2011 before and after the Tohoku-Oki earthquake, which occurred on March 11, 2011.

The horizontal polarization anomaly (HPA) of the surface waves, ($\psi_p$), for the entire period of the study is computed for each pair of receivers, through investigation of the off-diagonal components of the cross-correlation tensor of ambient seismic noise.

The HPA determines the orientation of the quasi-Rayleigh wave when seismic anisotropy cannot be neglected[14]. As a new observable, this is different from the seismic velocity change, as the HPA is related to the anisotropic property changes of the medium, but can also be affected by heterogeneities and the inhomogeneity of noise sources[11,12]. Indeed, temporal changes in the HPA can arise from seasonal variations or co-seismic changes in the seismic anisotropy. However, both of these changes do not follow the same temporal evolution that allows for the separation of one from the other. On a shorter time scale (i.e., of a few days), HPA variations might be due to changes in crack distributions due to local stress changes. This is the main focus of the present study.

Figure 2 shows that the largest and fastest change in the HPA during 2011 occurred at the time of the Tohoku-Oki earthquake (Supplementary Fig. 1). The procedure followed is similar to the method proposed by[11,12,15], and it is detailed in the "Methods" section. Note that the measured changes in the vertical polarization anomaly are much smaller than the changes in the HPA, and lie within the 2° error bar. This error is considered to be of the order of the HPA continuous fluctuations during calm periods (i.e., periods where no events or fast changes are observed; Fig. 2a).

The main HPA change appears to spread over almost 7 days, which corresponds to the length of the stacking window in the cross-correlation process. This co-seismic change might then be nearly instantaneous.

An unexpected observation is the sharp and rapid relaxation that occurs almost a month later, when no obvious seasonal changes are observed. Some earthquakes were triggered by the Tohoku-Oki earthquake[16] (Fig. 2b, Supplementary Fig. 2). The Shizuoka earthquake ($M_w = 5.9$; March 15, 2011) was the only significant one in the study area. However, over this time period when the strong changes occurred, the fast change in the HPA can be seen to be influenced by the Tohoku-Oki earthquake, but not by the local Shizuoka earthquake.

**Average and static anisotropy**. We use a regionalization technique (described in the "Methods" section) to represent the spatial distribution of the HPA and to extract the orientation of the anisotropy $\psi_\alpha$. Figure 3a shows the static orientation of the anisotropy and the amplitude of the static HPA in the study area (Methods, (Eq. (2)) obtained using the cross-correlation tensors averaged over the year 2011.

Figure 3a shows two areas where anisotropy is strong (dark blue; >20°) and that appear to be anti-correlated with the topography. For the eastern area, $\psi_\alpha$ is primarily in the WNW-ESE direction, in agreement with[17–20]. For the western area, $\psi_\alpha$ is North-South. In this area, the alignment of the volcanic fissures related to the stress field appears to be influenced by the subduction of the Philippine Sea plate. These results are consistent with the estimation of the regional compression in this zone by[19]. This indicates that the regional stress field in this area might be determined by subduction of both the Pacific plate and the Philippine Sea plate.

Close to the summit, the amplitude of the anisotropy is small. This might be due to the nonuniform and complex distribution of cracks in the area that are associated with the conic shape of the volcano, and/or to the local complex stress field due to the loading of the volcano[17,19,21]. Note that the orientation of $\psi_\alpha$ at almost 15 km from the summit (Fig. 3a, red lines) shows no overall direction, as it might have been filtered out by the regionalization method. This result is, however, in agreement with the shear-wave splitting results[19].

**Temporal change in anisotropy**. Figure 3b, c show the amplitude of the temporal change of the HPA in the study area (Eq. (3)) measured on a time window of 7 days, at the time of the earthquake (Fig. 3b) and one month after the earthquake (Fig. 3c). Figure 3b, c show the average orientation of the anisotropy for these 7 days.

Figure 3b shows that strong co-seismic changes in anisotropy are primarily located on the eastern side of the study area, where the static anisotropy is strong (Fig. 3a). In the darkest blue area, there is significant change in $\psi_\alpha$, and the alignment of the cracks appears to deviate from the NE-SW direction.

Our interpretation is that the change in crack alignment induces a change in $\psi_\alpha$. The strong change in HPA can reach 30° for the eastern side between Mount Fuji and the Hakone volcanic area (Fig. 3b), where the hydrothermal activity is high. As complementary information, strong velocity drops (0.1%) were observed for the date of the Tohoku-Oki earthquake not only in the area close to the epicenter, but also under the volcanic regions, such as the northern Honshu volcanic front and Mount Fuji[7]. We believe that these velocity decreases might be related to the opening of cracks[22].

After the Tohoku-Oki earthquake, the changes in dynamic stress due to the surface waves from the Tohoku-Oki earthquake contributed significantly to the initiation of the sequence of triggered seismic activity[23]. An increase in permeability, and hence of fluid flow on the fault, is directly related to the initiation of post-Tohoku-Oki earthquakes mainly beneath the Hakone volcano[24] (Fig. 2b).

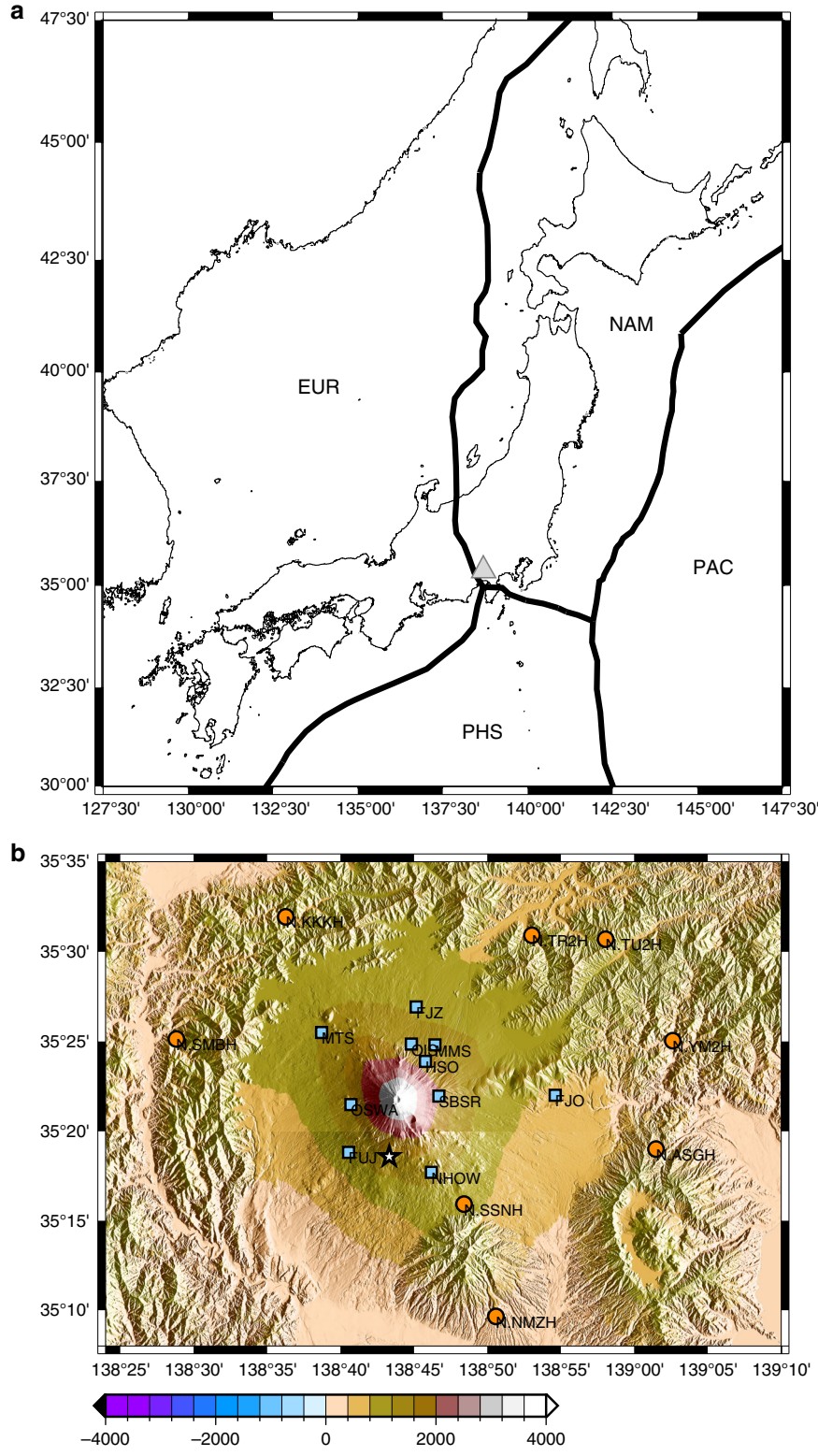

**Fig. 1** Map of the study area: **a** Tectonic setting around Mount Fuji. Thick black lines, plate boundaries of the Philippine Sea Plate, the American Plate, the Okhotsk, and the Pacific Plate; triangle, location of Mount Fuji; star, hypocenter of the Tohoku-Oki earthquake. **b** Map of Mount Fuji with topography. Blue squares, seismic stations installed by the Japan Meteorological Agency, the National Research Institute for Earth Science, and the Earthquake Research Institute; orange circles, Hi-net stations installed by the National Research Institute for Earth Science[13]; white star, epicenter of the Shizuoka earthquake; black square, meteorological station installed by the Hot Spring Research Institute of Kanagawa Prefecture. All of the seismic stations are equipped with three-component short-period sensors or broadband seismic sensors, with a sampling interval of 0.01 s.

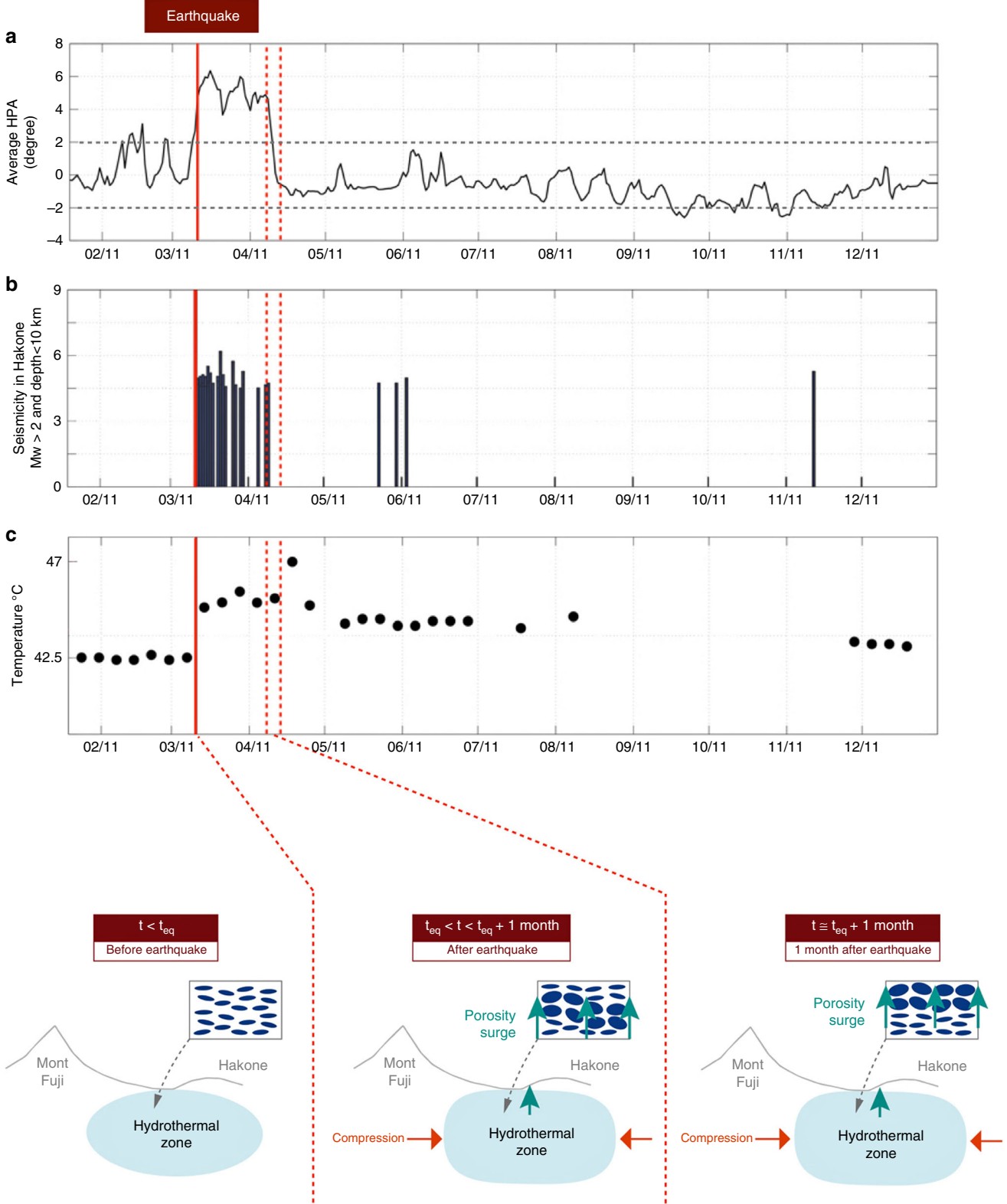

**Fig. 2** Temporal changes and association with the porosity surge: **a** Averages of the absolute values of the horizontal polarization anomaly (HPA) for all of the processed receiver pairs and through the year 2011. Note that the temporal resolution is 7 days (i.e., length of the stacking window of the cross-correlations) and the error on the HPA is of the order of 2°. **b** Seismicity in the study area. Vertical lines represent earthquakes with magnitudes >2 and depths <10 km. **c** Changes in temperature of the hot spring waters at station A (Fig. 1) in the Hakone area. Vertical red line, time of the Tohoku-Oki earthquake. Bottom: Scheme representing the propagation of the porosity surge in the hydrothermal system (blue) between the area East of Mount Fuji and beneath Mount Hakone. Small blue ellipses, fissures in the medium filled with hydrothermal fluids. The size of the ellipses increases during the passage of the hydrothermal fluid surge.

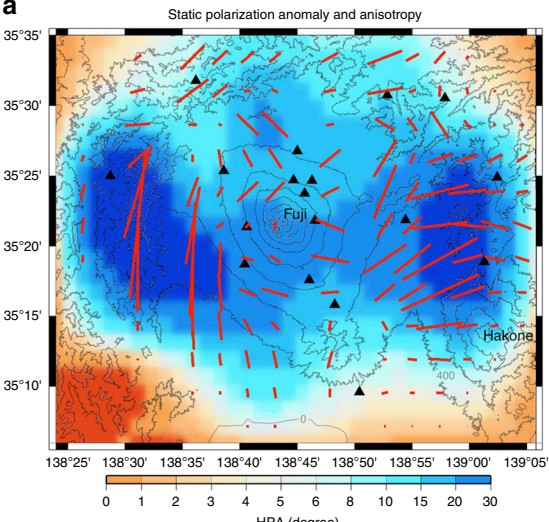

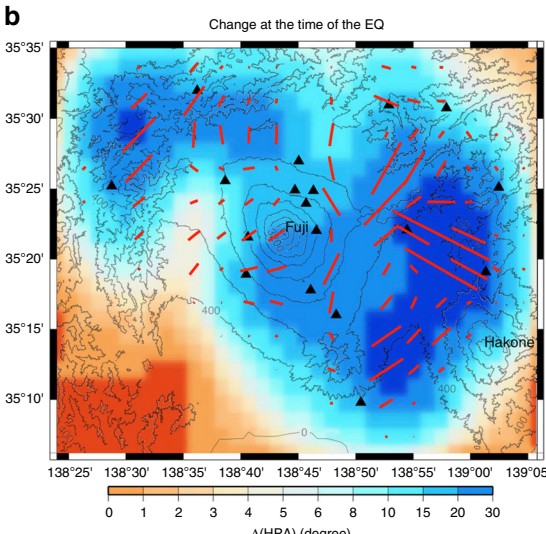

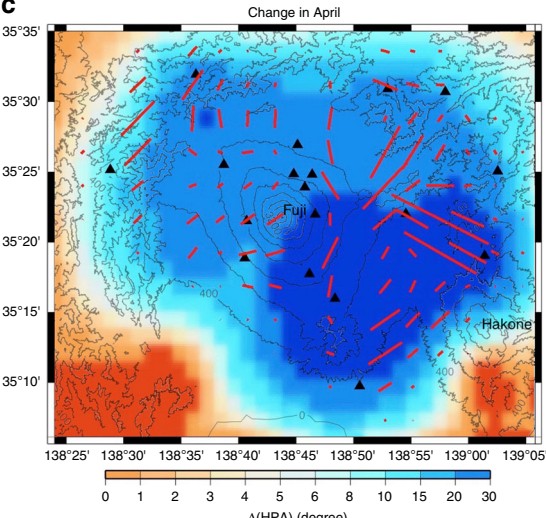

**Fig. 3** Regionalization of the horizontal polarization anomaly (HPA): **a** Map of the regionalized static anisotropy through 2011. Red lines, average orientation of static anisotropy. The lengths of the red lines and the color bar define the amplitudes of the static HPA through 2011. **b** Map of the regionalized co-seismic changes in anisotropy computed for a time window of 7 days centered on the date of the Tohoku-Oki earthquake. **c** Map of the regionalized sharp changes in anisotropy in April computed for a time window of 7 days, covering the decrease period of the HPA (Fig. 2a). Red lines, mid-orientation of anisotropy between before and after the changes. The lengths of the lines and the color bar define the amplitudes of the HPA temporal changes over the 7 days. **a–c** Black triangles, seismic stations; gray lines, topography contour lines. The regionalized orientation of the anisotropy is determined with a lateral resolution of ≈10 km.

amplitude of the change, the fast and strong decrease seems to be spread around the Hakone volcanic area. This unexpected observation might be due to a nonlinear dynamic effect of the hydrothermal system.

The seismic tomography study of ref. [25] for the magma-hydrothermal system beneath Hakone volcano shows the presence of cracks filled with water or carbon dioxide supplied from the inferred deep magma chamber.

Many previous studies have shown that hydrological responses to earthquakes are mainly triggered by changes in permeability, which, in turn, are a response to cyclic deformation and transient fluid flow[26]. The results here (Figs. 2, 3b) suggest that the passage of the seismic waves in the study area activated the hydrothermal system by shaking it, and caused percolation of the fluids trapped in this system. The result is a build-up of over-pressure in the upper crust, which produces an increase in the HPA.

If the change in HPA is associated with a pulse of fluid or pressure transferred from that depth to the surface, then the time scale of the process as constrained by the time evolution of the HPA is between 1 and 2 months. This gives an average velocity of ≈$10^{-3}$ ms$^{-1}$. Based on the end-member solutions for a porosity flow described in the "Methods" section, it appears that the propagation of a porosity surge in the hydrothermal system is either viscous or visco-elastic, and not a magmatic transfer.

We can furthermore consider as a rule of thumb that a three-dimensional porosity surge would correspond to an increase in the porosity of the medium by a factor of 2–8 (before returning to the nominal value once the solitons have passed)[27]. Based on the model of ref. [28], such an increase in porosity for a porous medium would induce a similar increase in anisotropy, which is consistent with the measured change in the anisotropy.

The surface temperature measurements for the hot springs in the Hakone area (Fig. 2c) show a significant co-seismic change (≈3 °C) that might be due to perturbation of the fluids close to the surface and the start of the purge from the deeper hydrothermal zone. The surface temperature measurements also show a transient and similar increase (≈2 °C) in April, slightly after the return to the original unperturbed seismic anisotropy state. Indeed, if the flow in the porous medium was a homogeneous Darcy's flow, it would be expected that the thermal anomaly measured in the springs will be maintained as long as the permeability remains open. However, because of the nonlinear relationship between porosity and permeability in porous systems, a homogeneous Darcy's flow is not stable and is indeed bound to produce porosity surges, possibly as solitary waves or porosity shocks, as a function of the effective rheology of the medium (i.e.,[29,30]). When such a surge reaches the surface, it will produce an extra increase in the temperature in the springs (here as an additional 2 °C thermal anomaly) because the flux of hot fluids increases (Fig. 2). We furthermore show that the velocity of a porosity surge is consistent with the 1-month delay between the

**Triggering of a hydrothermal fluid surge**. The regionalization of the HPA decrease in April, 2011 (Fig. 3c), shows that the mid-orientation of the anisotropy to the East of Mount Fuji is almost the same as for Fig. 3b, which corresponds to a return to the original unperturbed state of the fissure distribution. As for the

activation of the hydrothermal system and the thermal peak recorded in the springs. Finally, because the quantity of fluid is much larger in the surge than through the background flow, we consider that the surge extracts enough fluid from the system to go back to its initial stage, and hence for the return to the original seismic anisotropy state.

The fast decrease in the surface temperature is further evidence of the propagation of aqueous fluid and not magma, which cools more slowly.

We were faced with the interpretation of four different coeval phenomena 1 month after the Tohoku-Oki event; the fast decrease in the HPA, the end of the local seismicity, the variations in the temperature in the hot springs, and the changes in the orientation and amplitude of the anisotropy east of Mount Fuji, around the hydrothermal Hakone area. We believe that the interpretation in terms of a hydrothermal fluid surge is consistent with these different pieces of information.

## Discussion

Volcanic areas are sensitive to stress perturbations due to the presence of highly pressurized fluids in volcanic fissures. The magnitude of the HPA change makes it a reliable parameter to be used to image and characterize volcanic systems and the stress changes in the sensitive crust in these areas. The anisotropy measurements at Mount Fuji show a rapid co-seismic change at the time of the Tohoku-Oki earthquake, and a rapid and non-linear return to normal after 1 month, mainly beneath the Hakone area. A possible interpretation of these changes is the propagation of a porosity surge from the base of the hydro-thermal system up to the surface, which appears to be, to the best of our knowledge, the first seismic observational evidence of a hydrothermal fluid surge.

## Methods

**Monitoring surface-wave polarization by passive interferometry**. In an iso-tropic medium, the polarization of Rayleigh waves is within a plane that includes the vertical direction and the horizontal direction parallel to the propagation vector. The polarization of Love waves is along the transverse direction perpen-dicular to the propagation direction. However, in an anisotropic medium, there are deviations to the polarization that are associated with quasi-Rayleigh and quasi-Love waves[14]. This deviation is known as a polarization anomaly, and it can be measured using the signal on the off diagonal terms of a cross-correlation tensor (CCT) of ambient seismic noise: ZT, TZ, RT, and TR (i.e., refs. [11,12,15]).

The nine-component CCT is computed between the pairs of three-component (i.e., vertical Z, radial R, transverse T) seismograms and for a time window of 1 h (Supplementary Fig. 3). Each component of the CCT is normalized with respect to the total energy of the tensor (ref. [12]). The CCTs are then filtered within the frequency band of 0.12–0.25 Hz, which covers the most energetic secondary microseismic peak. To improve the signal-to-noise ratio and preserve temporal resolution, the CCTs are stacked over a moving time window of 7 days.

Following this procedure, which is similar to the method proposed by refs. [11,12,15] (Supplementary Fig. 4), the off-diagonal terms of the CCTs are investigated. If seismic sources are randomly distributed in the medium, these four off-diagonal components of the CCT are close to zero in an isotropic homogeneous medium. However, if the medium is anisotropic, the quasi-Rayleigh waves and quasi-Love waves generate significant signals on these four components, as expected from the deviated polarization of the surface waves.

An optimal rotation algorithm[31] is applied to the CCTs to minimize the off-diagonal components and separate the Rayleigh and Love waves. The optimal rotation algorithm extracts the set of angles $\delta_p$ and $\psi_p$ that minimizes the components ZT, TZ, RT, and TR of the original CCT. $\delta_p$ and $\psi_p$ are the vertical and horizontal polarization angles, which determine the deviation of the quasi-Rayleigh and quasi-Love waves[11,12].

For each pair of receivers, the HPA ($\psi_p$) of the quasi-Rayleigh wave is computed for the entire period of the study of almost 1 year (Supplementary Fig. 5).

**Anisotropy regionalization method**. A regionalization technique inverts the azimuthal anisotropy of surface waves into the spatial distribution of the polar-ization anomaly and the anisotropy direction $\psi_\alpha$ in the region of the study[12,32]. Based on the theory of ref. [33] and the numerical experiments of ref. [15], polarization of the fundamental mode of surface waves contains $2\psi$ and $4\psi$ azimuthal depen-dence. Hence, the polarization anomaly can be written to first order as a Fourier series, as follows:

$$\psi_p = a_1 \cos(2\psi) + b_1 \sin(2\psi) + a_2 \cos(4\psi) + b_2 \sin(4\psi), \quad (1)$$

where $\psi_p$ is the static horizontal polarization anomaly, $\psi$ is the azimuth of the pair of receivers relative to North, $a_n = \int_{-\pi}^{+\pi} \psi_p \cos(n\psi) d\psi$ and $b_n = \int_{-\pi}^{+\pi} \psi_p \sin(n\psi) d\psi$. Considering that the $2\psi$ terms are dominant for Rayleigh waves, $\psi_p$ can be written as follows:

$$\psi_p = \alpha \sin[2(\psi - \psi_\alpha)], \quad (2)$$

where $\psi_\alpha$ is the orientation of the anisotropy and $\alpha$ is the amplitude coefficient of $\psi_p$ that depends on the amplitude of anisotropy in the medium. $a_1 = -2\alpha \sin(2\psi_\alpha)$ and $b_1 = 2\alpha \cos(2\psi_\alpha)$. Hence, $\psi_\alpha = \frac{1}{2} \operatorname{atan}[-\frac{a_1}{b_1}]$.

The temporal variation of the horizontal polarization anomaly $\Delta\psi_p$ is obtained by taking the derivative of Eq. (2), knowing that the time-dependent parameters are $\psi_\alpha$, the orientation of anisotropy, and $\alpha$, which depends on the amplitude of the anisotropy. $\Delta\psi_p$ is then written as follows:

$$\Delta\psi_p = -2\alpha\Delta\psi_\alpha \cos[2(\psi - \psi_\alpha)] + \Delta\alpha \sin[2(\psi - \psi_\alpha)]. \quad (3)$$

Consequently, two possible effects can give rise to temporal variations in the horizontal polarization anomaly: a change in the orientation of the anisotropy $\Delta\psi_\alpha$, and/or a change in the amplitude $\Delta\alpha$ of the anisotropy.

For each horizontal polarization anomaly associated with a pair of receivers, we determine the orientation of the $2\psi$ anisotropy.

The sensitivity kernels of the noise cross-correlations can be considered as strongest near the two receivers. Consequently, the measurements are considered to be most sensitive to the structure around the two receivers. For every grid cell of $2.2 \times 2.2$ km, the average of the orientation of anisotropy is computed, and assigned with the corresponding weight for all of the paths that pass through the grid, considering that the weight of the measurement is highest near the receivers.

The regionalized orientation of the $2\psi$ anisotropy is then smoothed according to an empirical correlation length (here 5 km), as shown in Fig. 3a, which represents the static orientation of anisotropy (red lines) and the amplitude of the static polarization anomaly (length of red lines or colorbar) in the study region.

We selected almost 90% of the paths between the 18 stations, which covers the whole area.

We compute a prior model co-variance of the polarization anomaly between two stations A and B ([34]). Its formula is defined by $C_\psi(A, B) = \sigma_\psi(A)\sigma_\psi(B)e^{-\Delta^2(A,B)/2\lambda^2}$, where $\sigma_\psi(A)$ and $\sigma_\psi(B)$ are a-priori errors on the polarization anomalies at A and B, $\Delta(A, B)$ is the interstation distance, and $\lambda$ is the correlation length.

The last of these is chosen to be of the order of magnitude of the interstation distance and in a way to optimize the reduction from prior to posterior model variance. The interstation distances and the correlation lengths are chosen to be small (around 5 km), so we get good coherence between the two receivers, especially as the polarization measurement is not cumulative. In this way we measure local effects and avoid averaging different measurements that are influenced by different anisotropic areas.

The error on the polarization anomaly is set to 2° and the a-priori errors $\sigma_\psi(A)$ and $\sigma_\psi(B)$ are set to 25°. The choice of errors is dictated by what was observed in other tectonic contexts.

Our resultant model is the simplest model that can explain the data, and since the variance reduction for a correlation length of 5 km is relatively large (~90%), we can a-posteriori consider that this choice of correlation length was a reasonable assumption.

**Analytical solution to the hydrothermal fluid surge**. Fluid transfers in a hydrothermal system can be described as the flow of a visco-compressible fluid through a visco-elastic permeable matrix. Theoretical models of such flow predict a wide range of solutions as a function of the respective emphasis put on the con-servation of momentum, on the viscous deformation of the matrix[35], on its visco-elastic behavior[29] or on the compressibility of the fluid and/or of the matrix[30]. The relative importance of viscosity, visco-elasticity, and compressibility in the rheo-logical behavior of the system—hence on the velocity of the flow—can be estimated based on the Deborah number, $De$, defined as the ratio between the viscous time scale of compaction and the Maxwell relaxation-time,

$$De = \beta\Delta\rho g\delta, \quad (4)$$

where $\beta$ is the relevant compressibility (fluid or matrix), $\Delta\rho$ is the difference in density between the matrix and the fluid, $g$ is the acceleration of gravity, and $\delta$ is the compaction length defined as

$$\delta = \sqrt{\eta k_\phi/\mu}, \quad (5)$$

where $\eta$ is the bulk viscosity of the matrix, $\mu$ is the viscosity of the fluid, and $k_\phi$ is the permeability of the matrix, which is classically taken as

$$k_\phi = 10^{-3} a^2 \phi, \quad (6)$$

where $a$ is the particle radius, and $\phi$ is the porosity. As the rheological parameters are not well constrained in a volcanic or hydrothermal system at depth, it remains difficult to know precisely the value of $De$, and hence to obtain a complete solution for the porous flow. However, a diagnostic on the nature of the fluid and/or flow involved can be established by considering the two end-member flows that occur at very large and very small $De$. At small $De$, the flow is controlled by the balance of buoyancy with the bulk viscous resistance of the matrix to the deformation and the viscous shear of the fluid on the matrix, and tends to produce porosity surges known as solitons[27]. The speed of a soliton depends on the relationship between the permeability and the porosity, and is a fraction of the Darcy's velocity that corresponds to the maximum porosity in the soliton, $V_\phi$ (Supplementary Fig. 5),

$$V_\phi = \frac{k_0 \Delta \rho g}{\mu \phi_0}, \tag{7}$$

where $\phi_0$ is the maximum porosity in the soliton, and $k_0$ is the corresponding permeability. For a magmatic fluid, $a = 10^{-2}$ m, $\phi_0 = 0.1$, $\Delta \rho g = 500$ kg m$^{-1}$ and $\mu = 10^3$ Pa s, yield $V_\phi = 5 \times 10^{-9}$ m s$^{-1}$. For a hydrothermal fluid, taking $\Delta \rho g = 2000$ kg m$^{-3}$ and $\mu = 10^{-3}$ Pa s, this gives $V_\phi = 2 \times 10^{-3}$ m s$^{-1}$. At large $De$, the behavior of the system is controlled by its compressibility and production of poroelastic waves. An analytical expression for the speed of the porosity surge is ref. [36],

$$C_\phi \approx \frac{1 - \phi_0}{\phi_0} \frac{k_0 \Delta \rho g}{\beta \mu \sigma_0}, \tag{8}$$

where $\sigma_0$ is the effective stress in the hydrothermal system. Making the assumption that faults are preferred fluid permeation routes[37], and using the same parameters for the hydrothermal fault system as those used by ref. [36] for the Red Fault system, this gives $C_\phi \approx 5 \times 10^{-3}$ m s$^{-1}$, which is close to the previous estimation of the Darcy's velocity, a result confirmed by numerical resolution of the equations of compaction of a poro-viscoelastic medium at large $De$ (Supplementary Fig. 6).

## Data availability

The seismic data analyzed during the current study are available on the National Research Institute for Earth Science website: http://www.bosai.go.jp/e/activities/database/.

## Code availability

Computer codes and algorithms used in the current study are available upon request.

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

## Acknowledgements

The authors would like to thank the Japan Meteorological Agency, the Earthquake Research Institute, the National Research Institute for Earth Science, and the Hot Spring Research Institute of Kanagawa Prefecture for providing the data. This study was partly supported by LabEx UnivEarthS (ANR-11-IDEX-0005-02). Many thanks to the researchers who collaborated with the authors and contributed to the improvement of this paper, including to Martha Savage, Muriel Gerbault, Claude Jaupard, Steve Ingebritsen, and Jean-Philippe Metaxian. We would also like to thank Alexia Schroeder and Christopher Berrie for their editorial support. ISTerre is part of Labex OSUG@2020. Jean Paul Montagner acknowledges the support of the Institut Universitaire de France (CNRS, UMR 7154). Most of the computations were performed using the S-CAPAD platform of the Institut de Physique du Globe de Paris.

## Author contributions

J.P.M. and P.R. supervised the research project that was carried out by M.S., and provided, respectively, the regionalization code and the optimal rotation algorithm code. K. A. and F.B. provided the cross-correlations of continuous seismic data. E.K. worked on the interpretation of the data in terms of the propagation of a hydrothermal fluid surge. Y.A. provided the earthquake catalog and the temperature measurements in the study area. All authors contributed to the writing of the paper.

## Competing interests

The authors declare no competing interests.
