## [Peer Review File · Nature Communications]

Reviewers' comments:

Reviewer #1 (Remarks to the Author):

Summary:

The paper presents observations of time variability in the so-called horizontal polarisation anomaly (HPA) measured from the cross-terms (R-T correlations) of ambient noise cross-correlations in a volcanic area in Japan. The HPA is seen to change significantly in response to the 2011 Tohoku-Oki earthquake. This change is interpreted as evidence for activation of a hydrothermal system due to a shaking-induced permeability increase, which in turn triggers an upward-moving porosity wave. In addition to the seismological observations of the anisotropy increase and, in the following month, subsequent decrease, of the average HPA, this interpretation rests on the spatial distribution of the modified region, and the temperature of hot spring waters, which also shows a coseismic increase, and another positive blip shortly after anisotropy returns to pre-Tohoku values. The observation, if confirmed, is indeed rare and interesting, and would introduce another observable into volcano-seismology. These results therefore definitely are of wider importance and thus suitable for Nature Communications if they are robust.

However, the seismological results are somewhat poorly documented, so that it is hard to be fully confident about them (although some circumstantial evidence points to them indeed being robust, see more below). My main comment would therefore be that the whole processing chain needed for the derivation of the seismological results needs to be documented much more thoroughly and some possible sources of bias should be at least discussed, even if they ultimately can be ignored. (see below for detail).

I am not a hydrologist or expert in hydrothermal systems so feel less well qualified to comment on the proposed interpretation but I am under the impression that while the interpretation of the ascending porosity wave is certainly plausible, some more evidence would be required to make it compelling, and in particular I question the relevance of the temperature record. Some relatively moderate further analysis might be able to bolster (or refute) the interpretation, but as it stands it is somewhat speculative, in my opinion.

Because I expect the more detailed presentation and analysis to ultimately confirm the observations in the paper I fall on the side of "major revisions" (rather than reject and encourage to resubmit), but to make a fully convincing paper (to the high standards of Nature Comm.), some non-trivial additional analysis is probably required.

General comments:

* Documentation of the analysis

In order to really judge the quality of the data behind Fig. 2 and 3 you should provide additional information in the supplementary material

- Summarise how you calculate the cross-correlation functions (esp, what kind of normalisation do you use?)

- Show plots of RR, TT, and TR cross-correlation components for selected path(s) (7-day stack) for a time window with anomalous and normal HPA, ideally for one path across the region of most intense change, and one path not having experienced significant change. (For example, I expect the quality of the cross-correlation to deteriorate in the period immediately following the Tohoku earthquake due to intense aftershock activity; such plots would allow the reader to gauge the importance of this effect)

- The regionalisation scheme is not well explained. You only say that you calculate the average in each cell with the corresponding weight and that the sensitivity is larger close to the receivers. How do you actually distribute the sensitivity across your grid, though? Do you just use the great circle ray path between both receiver or do you assume an extended sensitivity region? What functional form does the sensitivity take to represent the higher sensitivity near the receivers? Also, the description seems to imply that the anisotropy model is derived following an approach similar to ART (Algebraic Reconstruction Technique) in early velocity tomography studies (but without iterating). Is there a reason why you don't set up and solve this as an inverse problem?

There is no discussion of the uncertainties of the 2D models shown in Fig. 2 either. What is the variance and resolution of these images? How well do they actually explain the data (i.e. the station-pairs HPA observations)? I think the most intuitive way to address this is by reporting data fit, and by carrying out synthetic reconstruction tests.

* In Saade et al (GJI 2015) you have very nicely tested the HPA predictions in velocity models without heterogeneity. But another possible way to get significant energy in the RT traces can be off-great circle path propagation arising from lateral isotropic heterogeneity. Volcanic areas tend to be quite heterogeneous, and from the map in Fig. 1 there seem to be small sedimentary basins surrounding Mt Fuji, so I would expect isotropic heterogeneity to be significant. You should estimate the likely magnitude of this effect and thus quantify the possible bias of the HPA inferred anisotropy. For the time variability I see less of a problem, as typical velocity changes are of the order of a few tenths of a percent, too small to significantly change ray paths.

* The period directly after the Tohoku earthquake is characterised by intense aftershock activity, which might have a strong influence on the quality of the cross-correlation functions, even when traces are normalised. Can you exclude any significant effect on the HPA?

* The coincidence of the occurrence of shallow seismicity sequence in the Hakone area in Fig 2c with the period of HPA change is really striking. While the onset is easily explained with the common trigger of the Tohoku earthquake, the end point of the sequence also seem to coincide with when the return of the HPA to pre-event levels. I was really surprised you did not discuss this coincidence, as I think this provides the most convincing circumstantial evidence for your interpretation, more so than the temperature curve. However, it is not really well described, where these earthquakes occur. So I suggest you plot these earthquakes also in map view. It might get a little busy, but I think you could still fit these into Figure 1, e.g. as small white circles (change the HiNet stations to squares in this case). If the selection area does not coincide with the map area, visualise this somehow, and/or plot shallow earthquakes outside the selection area with a different symbol/colour.

If there is really an upward propagating pulse, would one not expect a depth migration pattern in these earthquakes? This would be quite easy to check, and if there is depth progression, it would bolster your story considerably.

* On the interpretation: I am struggling a little to understand the significance of the temperature curve. According to the model in Fig 4, the porosity is increased initially at depth and only then propagates upward. So why would I expect an immediate temperature increase after the Tohoku EQ. Also, the measurement point is at the edge of the measurement area, i.e. at the very edge of the maps in Fig. 3, and actually in an area which did not show any anisotropy change (although I guess this might be due to coverage resp. resolution issues rather than the absence of anisotropy; here a better discussion of resolution would have been very helpful). So I wonder if this temperature record is not a 'red herring'. Is it possible to use a few additional HiNet stations to the east to improve the resolution at the site of the sampled hot springs and be able to state whether or not it is in an area having experienced coseismic HPA change. I am also wondering how to square the essentially boxcar-shaped HPA change curve in Fig 2a with the gradual upwards propagation. As a relatively narrow period band was used for analysis, I would have expected the magnitude of the anomaly to change gradually as the porosity moves up through the sensitivity region of the surface wave. It is probably not possible to measure the HPA over multiple bands due to the signal quality, but if it were possible, and you could see the HPA change move progressively to shorter periods, this would of course strongly bolster your interpretation.

Detailed comments

Abstract: I 22-28 In the abstract, a distinction is made between the coseismic change of HPA (due to stress perturbation), and the decrease one month later associated with the propagation of the hydrothermal fluid surge at the surface. Here my question would be if the 'stress perturbation' is static (i.e. the long term shift in stress due to displacement along the plate interface) or dynamic (during the passing of the seismic waves radiated from the earthquake)?

As far as I know, usually dynamic triggering is thought to be more important for volcanic systems, but it means that they really need to work by triggering a process with longer term implications, as the actual passage of the waves last only a relatively short time. As described in the abstract it sounds almost like a coincidence that the HPA returns back to almost exactly the original state after one month, as the two parts are described in the abstract as independent events (i.e. coseismic perturbation and arrival of fluid surge at the surface). From the discussion I gather that it is really the porosity created coseismically at depth then migrates upwards, and when the porosity wave has left the system, everything is naturally back to the original state – this needs to be clarified in the abstract also.

p 3

I 60 You should define briefly what the horizontal polarization anomaly represents, as the text should be understandable without referring to the supplementary material or additional papers

I 68-69 “The measured changes in the vertical polarization anomaly are much smaller

than changes in the HPA, and lie within the HPA error bars.” – do you mean VPA error bars? They seem more relevant here. Also, it would be interesting for the reader to judge for himself these anomalies, and I suggest adding a figure similar to Fig. 2a but for VPA to the supplementary material.

I. 72-72 “.. extends over almost 7 days (the length of the stacking window).” - I guess the implication here is that a step change could be smeared out because of the stacking window. You can spell this out more clearly, e.g. “appears to extend over almost 7 days. As this is the length of the stacking window, the change might be nearly instantaneous”.

I. 75-76 You mention the Shizuoka EQ, also shown in Fig. 1. I think it would be worthwhile to also mark this in Figure 2. Also you claim that the HPA is only influenced by the Tohoku EQ, not the Shizuoka one. But as these are just 4 days apart, and the stacking window is 7 days, can you really be confident that there is no discernable contribution from the Shizuoka EQ (I concede it cannot really be the dominant one)

p 5 | 155 ... and a rapid TRANSIENT increase in temperature ...

I 158 With fast decrease you presumably mean the quick reversal of the transient increase (in mid-April) as the coseismic step change only decays very slowly, and even at the end of your study period had not quite returned to pre-Tohoku levels.

Discussion

Although you mention stress quite often in the text, actually in the discussion you do not directly invoke coseismic changes to the stress field to explain your sudden shift in fast anisotropy directions

(unlike in the abstract). Instead you talk about the permeability, which is created. You should adapt abstract and/or discussion to achieve consistency.

Conclusion:

This reads more like an advertisement for your results rather than a summary of the results themselves, so seems more suitable for the cover letter than the conclusion. Just focus on your observations and their interpretation.

Fig.1

The inset map covers up too much of the map unnecessarily. By moving the colour bar to the right, and then shifting the inset left and down, I imagine you can get back two thirds of the map space now covered without increasing the footprint on the page.

Fig. 2

All: Why don't you show the data for the whole time you processed, i.e. starting earlier. The signal is quite suggestive, but there are also some (apparent?) HPA changes almost reaching the 2-sigma level in February, and it left me wondering how frequent such excursions are, and how significant the one observed after the Tohoku earthquake actually is.

On a purely presentational note, please label the full date for the x-axis notations (i.e. 1/2/2018 etc) as it is otherwise not clear if the ticks represent the beginning or middle of the month (of course, one can figure it out from the well-known date of the Tohoku earthquake)

(a) By showing the average HPA change, do you not make the change look less significant than it really is, as different parts of the map are not equally strongly affected or even necessarily in the same direction. I could imagine if you showed the HPA change for a single station-

pair with a path through the most strongly affected region, you would potentially see a stronger effect.

(d) The sampling of the temperature curve is really low, only about once per week, and with many gaps. Is this really the best time series available for the temperature. It would in particular be nice to get a few more samples next to the peak value to get an idea of the time scale of this excursion (and exclude a fluke measurement).

Caption/(a) You should mention the averaging length (7 days) and maybe mark this in a. Currently, it looks like the HPA change begins before the Tohoku earthquake (precursory), but I presume this is simply the effect of the centred averaging window)

Fig. 3

Seeing the change image is sometimes useful, but I would find it very helpful to also see the before and after situation. So instead of Fig. 3a (which I understand to show the situation averaged over the whole year), make a figure showing the 'before situation', i.e. the average anisotropy before March 11, 2011, and an 'after situation, i.e. the anisotropy averaged over the one month where it is anomalous. Depending how similar it is to the initial situation, you might wish to show another frame showing the situation after a return of the average HPA to pre-event values. If space is tight, the before and after images would be more useful than the change images (as the change seems to be quite large), but really it would be best to show both, and you would only need one more panel (which you could then arrange in a 2x2 configuration)

Some more detailed comments on this figure:

- (a)-(c) Is the legend for the color scale mislabeled. It looks like the colour is chosen according to the anisotropy amplitude, i.e. α (and not the average fast direction as is indicated, i.e. Ψ_a).
- Also, you explain the meaning of Ψ_a (and α) only in the supplementary material, so should choose a label here that does not require reading the appendix.
- In (b) and (c) I did not really understand if the colour relates to the absolute anisotropy amplitude or to how much the anisotropy changed.
- this might be subjective but personally I use red-white(pale)-blue colour scales primarily for situations where the central (white colours) are associated with zero perturbations, and find the use here, with red actually representing no anisotropy. I recommend to use a colour scale where zero corresponds to a neutral colour (black, white or grey).

Supplementary material

p 15 | 345-347 This sentence is strangely vague. What functional form was used for the kernel (e.g. as a function of distance from source and receiver point).

Language comments:

p 2

l 13 sensitive parameters -> parameters sensitive

l 17 Delete "trending" (it's just a filler word)

ambient -> ambient

l 24 event thought -> even though

l 35 Delete "in the media" (redundant, provides no useful information)

p 3

l 42 The question that remains -> An open question

l 46 reduction -> reductions (or change verb to "was")

l 47 where -> were

l 49 Consequently -> Here

l 57,58 recorded for the year 2011 -> for 2011

p 4

l 113 is related -> may be related

p 5

l 155 co-seismic significant -> significant coseismic

References:

Pay attention to capitalisation of place names throughout (if this is related to Bibtex, usage, just use an extra pairs of {} around proper nouns, or the whole title to preserve capitalisation)

P 16 l 402 Delete "reasonable"

Reviewer #2 (Remarks to the Author):

Comments

I recommend publication of the paper.

The authors found anisotropy changes around Mt Fuji after the Tohoku earthquake using noise crosscorrelation tensor (CCT). The method used was based on their early work on the ORA. Overall, I think the methodology is sound. The results are interesting and important. The paper is very terse. I recommend the authors to answer the following questions.

1. Line1: delete "first". I think it's too certain to call this first evidence of hydrothermal reactivation. Hydrothermal reactivation is the "interpretation" for your observation.
2. Shorten the 'abstract'. Delete first 2 sentences (e.g., in line10-14) and delete "for instance" in the 3rd sentence. Delete "therefore" in line16. Delete the sentence around line 21. These changes can make your abstract direct to the point.
3. Can you provide HPA error bars to fig2a? you mentioned this in line 68-69 but didn't show the error bars.
4. fig2a: If you do 5-day or 10-day averaging of CCT to get HPA? Will that change the HPA results?
5. Fig3a,b,c: the magnitude of the anisotropy (i.e., length of the red lines) is missing.
6. Fig3a,b,c: attach error bars for the Φ_{α} , i.e., a rose diagram of all possible directions.
7. Line77: what's the meaning of the word "period" here?
8. Line 81: in your regionalization method, did you attribute the HPA at the midpoint between the station pair?
9. Line87: How does the surface topography around the mountain affect the HPA?
10. Line 110: change "time" to "date".
11. Line115-116: what evidence supports this argument?
12. Line 118: how much dynamic stress has been changed? Authors should talk about the magnitude of the dynamic stress if this is a reason.
13. Line 126: define "mid-orientation"
14. Line 388: I think porosity $\phi_0=0.1$ is pretty large, i.e., a 1m cube of rock contains a 0.5m cube fluid? $0.5^3/1^3 \sim 0.1$. is this reasonable?

END

Reviewer #3 (Remarks to the Author):

The authors report on results of changes in seismic anisotropy after the 2011 Tohoku earthquake in the Mt. Fushi area, Japan. Anisotropy is determined by seismic noise cross-correlation. The authors interpret their results as changes in crack alignment that was due to a 'porosity surge' triggered by seismic waves from the Tohoku event.

The study could be of broad interest and is relatively well structured and written. However, I have significant concerns about results and interpretation. The observational evidence for the presented model is, in my opinion, thin and the far-reaching interpretations may not be supported by the data. The authors focus on horizontal polarization anomalies and modest temperature increase, to infer a vertical 'porosity surge' that initiated from a deep magma chamber and strongly changed crack alignments within the affected rock volume. The observations do not unequivocally support this conclusion and no other mechanisms are discussed. A plausible explanation for the observations is simply the mobilization of near surface fluids due to large co-seismic ground motions, resulting in temperature increase. Note that the apparent onset of anisotropy change and temperature increase seem to occur at the same time which would not be expected for a deep source initiating from the magma chamber. Changes in anisotropy may originate from a variety of sources including stress induced anisotropy and fabric related anisotropy. It is difficult to assess the validity of the proposed mechanism in the absence of stronger observational evidence.

Below are the point-by-point responses to the comments of the reviewers:

Reviewer #1:

Summary:

The paper presents observations of time variability in the so-called horizontal polarisation anomaly (HPA) measured from the cross-terms (R-T correlations) of ambient noise cross-correlations in a volcanic area in Japan. The HPA is seen to change significantly in response to the 2011 Tohoku-Oki earthquake. This change is interpreted as evidence for activation of a hydrothermal system due to a shaking-induced permeability increase, which in turn triggers an upward-moving porosity wave. In addition to the seismological observations of the anisotropy increase and, in the following month, subsequent decrease, of the average HPA, this interpretation rests on the spatial distribution of the modified region, and the temperature of hot spring waters, which also shows a coseismic increase, and another positive blip shortly after anisotropy returns to pre-Tohoku values. The observation, if confirmed, is indeed rare and interesting, and would introduce another observable into volcano-seismology. These results therefore definitely are of wider importance and thus suitable for Nature Communications if they are robust. However, the seismological results are somewhat poorly documented, so that it is hard to be fully confident about them (although some circumstantial evidence points to them indeed being robust, see more below). My main comment would therefore be that the whole processing chain needed for the derivation of the seismological results needs to be documented much more thoroughly and some possible sources of bias should be at least discussed, even if they ultimately can be ignored. (see below for detail).

I am not a hydrologist or expert in hydrothermal systems so feel less well qualified to comment on the proposed interpretation but I am under the impression that while the interpretation of the ascending porosity wave is certainly plausible, some more evidence would be required to make it compelling, and in particular I question the relevance of the temperature record. Some relatively moderate further analysis might be able to bolster (or refute) the interpretation, but as it stands it is somewhat speculative, in my opinion.

Because I expect the more detailed presentation and analysis to ultimately confirm the observations in the paper I fall on the side of “major revisions” (rather than reject and encourage to resubmit), but to make a fully convincing paper (to the high standards of Nature Comm.), some non-trivial additional analysis is probably required.

General comments:

* Documentation of the analysis

In order to really judge the quality of the data behind Fig. 2 and 3 you should provide additional information in the supplementary material.

We added the complete workflow in the supplementary material (SM). More technical details can be found in the previous papers (Saadé et al., GJI, 2018).

- Summarize how you calculate the cross-correlation functions (esp, what kind of normalisation do you use?) We normalise each correlation tensor by its total energy. This information is added to the text.

- Show plots of RR, TT, and TR cross-correlation components for selected path(s) (7-day stack) for a time window with anomalous and normal HPA, ideally for one path across the region of most intense change, and one path not having experienced significant change. (For example, I expect the quality of the cross-correlation to deteriorate in the period immediately following the Tohoku earthquake due to intense aftershock activity; such plots would allow the reader to gauge the importance of this effect)

As requested, we present in the SM example of cross-correlogram. We do not see a clear effect of the aftershocks on the cross-correlations, in the frequency range we consider.

- The regionalization scheme is not well explained. You only say that you calculate the average in each cell with the corresponding weight and that the sensitivity is larger close to the receivers. How do you actually distribute the sensitivity across your grid, though? Do you just use the great circle ray path between both receiver or do you assume an extended sensitivity region? What functional form does the sensitivity take to represent the higher sensitivity near the receivers? Also, the description seems to imply that the anisotropy model is derived following an approach similar to ART (Algebraic Reconstruction Technique) in early velocity tomography studies (but without iterating). Is there a reason why you don't set up and solve this as an inverse problem?

The regionalisation technique is similar to the method for inverting azimuthal anisotropy of surface waves described in (Montagner & Nataf 1986). It based on a gaussian distribution with a correlation length Λ_{cor} . (see the previous paper, Saade et al., GJI, 2018)

The latter does not depend on the inter-station distance. In fact, the particularity of anisotropy and the measurement of its orientation is that we cannot locate it between the stations. We do not have access to the limits of the anisotropic area between the two receivers when only using the polarization data.

Hence, we explored different values for the correlation length Λ_{cor} in order to determine the Λ_{cor} associated with the minimum computed variance of residuals (Tarantola and Valette 1982). The maximum sensitivity is around the receivers and the value of the optimal correlation length we found is 5km (of the order of the average inter-station distance). Note that for the area of Iwate-Miyagi (Saade et al. 2018) the correlation length was 8km. This is why a dense network and using small

inter-station distances is essential in order to locate the anisotropy variations with the best resolution.

There is no discussion of the uncertainties of the 2D models shown in Fig. 2 either. What is the variance and resolution of these images? How well do they actually explain the data (i.e. the station-pairs HPA observations)? I think the most intuitive way to address this is by reporting data fit, and by carrying out synthetic reconstruction tests.

The covariance reduction of the regionalisation is of the order of 90% (its formula is added in the SM) which is quite low and the lateral resolution is 2.2x2.2 km (which is the size of the lateral grids). We also selected 139 out 153 paths between the stations which covers the whole area. These infos are added to the text.

* In Saade et al (GJI 2015) you have very nicely tested the HPA predictions in velocity models without heterogeneity. But another possible way to get significant energy in the RT traces can be off-great circle path propagation arising from lateral isotropic heterogeneity. Volcanic areas tend to be quite heterogeneous, and from the map in Fig. 1 there seem to be small sedimentary basins surrounding Mt Fuji, so I would expect isotropic heterogeneity to be significant. You should estimate the likely magnitude of this effect and thus quantify the possible bias of the HPA inferred anisotropy. For the time variability I see less of a problem, as typical velocity changes are of the order of a few tenths of a percent, too small to significantly change ray paths.

Indeed lateral heterogeneities can deviate the polarization of surface waves. This is why we select pairs of stations with the minimised inter-station distance in order to consider the hypothesis that the medium between the stations is homogeneous. For instance, the average inter-station distance considered in this study is 5 km.

Besides, in the study we focus on temporal changes of HPA that are related to anisotropy rather than lateral heterogeneities.

* The period directly after the Tohoku earthquake is characterized by intense aftershock activity, which might have a strong influence on the quality of the cross-correlation functions, even when traces are normalized. Can you exclude any significant effect on the HPA?

The aftershocks in the rupture area of the Tohoku-Oki earthquake are smaller than the main earthquake and are quite far, so they cannot have the same influence on the area of Mount Fuji as the more local microseismic activity. Besides the aftershock activity in the Tohoku region did not drop suddenly one month after the mainshock.

Nonetheless, aftershocks can be considered as additional source of noise and can only increase the signal of the CCTs but they do not change the value of the measured HPA. Especially when the sources of aftershocks are quite far from the network of stations at Mt Fuji and so the source incidence is almost similar at all stations.

Most importantly, if we look at the cross-correlograms (added figures in the SM), we do not see the effect of the aftershocks on the filtered cross-correlations.

* The coincidence of the occurrence of shallow seismicity sequence in the Hakone area in Fig 2c with the period of HPA change is really striking. While the onset is easily explained with the common trigger of the Tohoku earthquake, the end point of the sequence also seem to coincide with when the return of the HPA to pre-event levels. I was really surprised you did not discuss this coincidence, as I think this provides the most convincing circumstantial evidence for your interpretation, more so than the temperature curve. However, it is not really well described, where these earthquakes occur. So I suggest you plot these earthquakes also in map view. It might get a little busy, but I think you could still fit these into Figure 1, e.g. as small white circles (change the HiNet stations to squares in this case). If the selection area does not coincide with the map area, visualise this somehow, and/or plot shallow earthquakes outside the selection area with a different symbol/colour. If there is really an upward propagating pulse, would one not expect a depth migration pattern in these earthquakes? This would be quite easy to check, and if there is depth progression, it would bolster your story considerably.

Thank you for pointing out this idea.

The aftershocks represented in figure 2c, are located in the Hakone area only (almost 30x30 km²).

We plotted the migration to the surface of the aftershocks in the area of Hakone (figure below). But we do not see a clear migration of the aftershocks with depth. The main observation is the abundance of aftershocks during almost one month after the main shock.

* On the interpretation: I am struggling a little to understand the significance of the temperature curve. According to the model in Fig 4, the porosity is increased initially at depth and only then propagates upward. So why would I expect an immediate temperature increase after the Tohoku EQ. Also, the measurement point is at the edge of the measurement area, i.e. at the very edge of the maps in Fig. 3, and actually in an area which did not show any anisotropy change (although I guess this might be due to coverage resp. resolution issues rather than the absence of anisotropy; here a better discussion of resolution would have been very helpful). So

I wonder if this temperature record is not a 'red herring'. Is it possible to use a few additional HiNet stations to the east to improve the resolution at the site of the sampled hot springs and be able to state whether or not it is in an area having experienced coseismic HPA change. I am also wondering how to square the essentially boxcar-shaped HPA change curve in Fig 2a with the gradual upwards propagation. As a relatively narrow period band was used for analysis, I would have expected the magnitude of the anomaly to change gradually as the porosity moves up through the sensitivity region of the surface wave. It is probably not possible to measure the HPA over multiple bands due to the signal quality, but if it were possible, and you could see the HPA change move progressively to shorter periods, this would of course strongly bolster your interpretation.

Unfortunately we don't have access to more data from more seismic stations, and the procedure of getting the data, converting and processing it is quite long, and beyond the scope of this paper.

Indeed, the frequency range should be chosen precisely, it should cover one micro-seismic peak and give good quality cross-correlations from which we have a minimized misfit at the rotation. The selected frequency range is relatively narrow, so we have no depth resolution. Besides that, the HPA measurements are not cumulative, as the velocity measurements. Having a gradual measurement with depth is not trivial.

Concerning the temperature measurement, there are **two** main effects giving rise to a temperature sudden change: 1- the co-seismic temperature increase due to the perturbation of the fluids close to or on the surface and the start of the purge from the deeper hydrothermal zone, 2- the complete purge of the system and the arrival at the surface of all the fluids that fill the cracks in the medium.

Note that, the return to normal of the surface temperature is slower than that of anisotropy, since the temperature diffusion is slower.

Detailed comments

Abstract: I 22-28 In the abstract, a distinction is made between the coseismic change of HPA (due to stress perturbation), and the decrease one month later associated with the propagation of the hydrothermal fluid surge at the surface. Here my question would be if the 'stress perturbation' is static (i.e. the long term shift in stress due to displacement along the plate interface) or dynamic (during the passing of the seismic waves radiated from the earthquake)? There is the static HPA induced by the static stress in the area (Fig 3a) and the temporal change of HPA (change relatively to the static HPA and then return to normal, here after one month) induced by the "dynamic" stress perturbation (Fig 3b,c). Actually, it is a quite difficult question, since we average all physical properties over one week so we have no direct access to the time scale of the propagating waves. So there are two effects of Japan-Tohoku earthquake on the stress field: seismic waves induced a dynamic change in the stress field triggering the porosity wave and on a longer time basis the influence of the static stress field.

As far as I know, usually dynamic triggering is thought to be more important for volcanic systems, but it means that they really need to work by triggering a process with longer term implications, as the actual passage of the waves last only a relatively short time. As described in the abstract it sounds almost like a coincidence that the HPA returns back to almost exactly the original state after one month, as the two parts are described in the abstract as independent events (i.e. coseismic perturbation and arrival of fluid surge at the surface). From the discussion I gather that it is really the porosity created coseismically at depth then migrates upwards, and when the porosity wave has left the system, everything is naturally back to the original state – this needs to be clarified in the abstract also.

The stress perturbation is at the origin of the triggered porosity surge, but it's the porosity surge that changed the anisotropy in the medium. The time of the HPA change is then influenced by the time of the porosity surge. We made this point clearer in the abstract.

p 3

I 60 You should define briefly what the horizontal polarization anomaly represents, as the text should be understandable without referring to the supplementary material or additional papers

More details were added to the text in red

I 68-69 “The measured changes in the vertical polarization anomaly are much smaller than changes in the HPA, and lie within the HPA error bars.” – do you mean VPA error bars? They seem more relevant here. Also, it would be interesting for the reader to judge for himself these anomalies, and I suggest adding a figure similar to Fig. 2a but for VPA to the supplementary material.

It is possible to regionalise the VPA measurements, but they will not provide any interesting information since they are practically zero. They “lie within the HPA error bars” which means they are of the order of the error on the measured angle by the ORA. In the previous applications of this method (Parkfield earthquake Durand et al.2011, Iwate-Miyagi Saade et al.2017) it was also observed that the VPA measurements were negligible, which is not surprising since the perturbations of the stress field are mainly horizontal.

I. 72-72 “.. extends over almost 7 days (the length of the stacking window).” - I guess the implication here is that a step change could be smeared out because of the stacking window. You can spell this out more clearly, e.g. “appears to extend over almost 7 days. As this is the length of the stacking window, the change might be nearly instantaneous”. Done

I. 75-76 You mention the Shizuoka EQ, also shown in Fig. 1. I think it would be worthwhile to also mark this in Figure 2. Also you claim that the HPA is only influenced by the Tohoku EQ, not the Shizuoka one. But as these are just 4 days apart, and the stacking window is 7 days, can you really be confident that there is no discernable contribution from the Shizuoka EQ (I concede it cannot really be the dominant one). We explored the period of the earthquakes (Tohoku and Shizuoka) closely and it seems that the strong “coseismic” HPA change starts before the time of the Shizuoka is included in the stacking window of 7 days. I hope this clarifies your point.

Figure: Average of the horizontal polarization anomaly (HPA) zoomed around the period where the strong changes occurred. Top panel: in red is the period where changes of HPA can be influenced by the Tohoku earthquake (March 11). Bottom panel: in red is the period where the changes of HPA can be influenced by the Shizuoka earthquake (March 15). Note that the length of the moving stack window is 7 days.

p 5 | 155 ... and a rapid TRANSIENT increase in temperature ... **Corrected**

l 158 With fast decrease you presumably mean the quick reversal of the transient increase (in mid-April) as the coseismic step change only decays very slowly, and even at the end of your study period had not quite returned to pre-Tohoku levels.

As we mentioned earlier the anisotropy change is mainly influenced by the porosity surge which has a non-linear behavior.

Discussion

Although you mention stress quite often in the text, actually in the discussion you do not directly invoke coseismic changes to the stress field to explain your sudden shift in fast anisotropy directions (unlike in the abstract). Instead you talk about the permeability, which is created. You should adapt abstract and/or discussion to achieve consistency.

Seismic anisotropy can be induced by both effects, stress perturbation (inducing change in the orientation of anisotropy) and/or permeability change (inducing change in the amplitude of anisotropy). We focus on the permeability change since it seems to be the dominant phenomenon affecting seismic anisotropy in the area (which could explain the rapid « non-linear » decrease of the HPA one month after the earthquake). But we added some details in the discussion about the stress induced anisotropy change in order to make this point clearer.

Conclusion:

This reads more like an advertisement for your results rather than a summary of the results themselves, so seems more suitable for the cover letter than the conclusion. Just focus on your observations and their interpretation.

The conclusion is modified accordingly.

Fig.1

The inset map covers up too much of the map unnecessarily. By moving the colour bar to the right, and then shifting the inset left and down, I imagine you can get back two thirds of the map space now covered without increasing the footprint on the page.

Fig. 2

All: Why don't you show the data for the whole time you processed, i.e. starting earlier. The signal is quite suggestive, but there are also some (apparent?) HPA changes almost reaching the 2-sigma level in February, and it left me wondering how frequent such excursions are, and how significant the one observed after the Tohoku earthquake actually is.

The fluctuations (for instance, in February) can be related to local effects, such as seasonal changes. But the important observation is that the co-seismic change is quite spectacular and occurs at the time of the Tohoku-Oki earthquake.

On a purely presentational note, please label the full date for the x-axis notations (i.e. 1/2/2018 etc) as it is otherwise not clear if the ticks represent the beginning or middle of the month (of course, one can figure it out from the well-known date of the Tohoku earthquake)

(a) By showing the average HPA change, do you not make the change look less significant than it really is, as different parts of the map are not equally strongly affected or even necessarily in the same direction. I could imagine if you showed the HPA change for a single station-pair with a path through the most strongly affected region, you would potentially see a stronger effect.

I added a figure in the supplementary materials showing the HPA change for different pairs of receivers (where we observe strong changes and small changes).

(d) The sampling of the temperature curve is really low, only about once per week, and with many gaps. Is this really the best time series available for the temperature. It would in particular be nice to get a few more samples next to the peak value to get an idea of the time scale of this excursion (and exclude a fluke measurement).

Unfortunately these are the only temperature data available with a good signal to noise ratio. We had 3 other surface temperature measurements but they were not localised far from the Hakone area and have very noisy records (see Itadera et al. 2011).

Caption/(a) You should mention the averaging length (7 days) and maybe mark this in a. Currently, it looks like the HPA change begins before the Tohoku earthquake (precursory), but I presume this is simply the effect of the centered averaging window) Exactly, I added this info in the caption of the figure.

Fig. 3

Seeing the change image is sometimes useful, but I would find it very helpful to also see the before and after situation. So instead of Fig. 3a (which I understand to show the situation averaged over the whole year), make a figure showing the 'before situation', i.e. the average anisotropy before March 11, 2011, and an 'after situation', i.e. the anisotropy averaged over the one month where it is anomalous. Depending how similar it is to the initial situation, you might wish to show another

frame showing the situation after a return of the average HPA to pre-event values. If space is tight, the before and after images would be more useful than the change images (as the change seems to be quite large), but really it would be best to show both, and you would only need one more panel (which you could then arrange in a 2x2 configuration)

We only have less than 2 months of measurements before the Tohoku-Oki earthquake, where we see a lot of fluctuations in the HPA measurements. A regionalisation of the static anisotropy in the period would not be reliable.

Some more detailed comments on this figure:

- (a)-(c) Is the legend for the color scale mislabeled. It looks like the color is chosen according to the anisotropy amplitude, i.e. α (and not the average fast direction as is indicated, i.e. Ψ_a). The color bar represents the values of the HPA. Thank you, we corrected the labels in the figures.

- Also, you explain the meaning of Ψ_a (and α) only in the supplementary material, so should choose a label here that does not require reading the appendix. Done, we used "HPA" instead.

- In (b) and (c) I did not really understand if the color relates to the absolute anisotropy amplitude or to how much the anisotropy changed. The colorbar relates to the HPA values. As indicated in the legend of the figure, in (a) it is related to the static HPA value, in (b) and (c) it is related to the HPA temporal change over 7 days.

- this might be subjective but personally I use red-white(pale)-blue color scales primarily for situations where the central (white colors) are associated with zero perturbations, and find the use here, with red actually representing no anisotropy. I recommend to use a color scale where zero corresponds to a neutral color (black, white or grey).

Supplementary material

p 15 | 345-347 This sentence is strangely vague. What functional form was used for the kernel (e.g. as a function of distance from source and receiver point).

A gaussian distribution with a correlation length as explained previously. We clarified this information in the MS.

Language comments:

p 2

l 13 sensitive parameters -> parameters sensitive Done

l 17 Delete "trending" (it's just a filler word) Done

ambient -> ambient Done

l 24 event thought -> even though Done

l 35 Delete "in the media" (redundant, provides no useful information) Done

p 3

l 42 The question that remains -> An open question Done

l 46 reduction -> reductions (or change verb to "was") Done

l 47 where -> were **Done**
l 49 Consequently -> Here **Done**
l 57,58 recorded for the year 2011 -> for 2011 **Done**

p 4

l 113 is related -> may be related **Done**

p 5

l 155 co-seismic significant -> significant coseismic **Done**

References:

Pay attention to capitalisation of place names throughout (if this is related to Bibtex, usage, just use an extra pairs of {} around proper nouns, or the whole title to preserve capitalization **Done**

P 16 l 402 Delete "reasonable" **Done**

Reviewer #2:

Comments

I recommend publication of the paper.

The authors found anisotropy changes around Mt Fuji after the Tohoku earthquake using noise crosscorrelation tensor (CCT). The method used was based on their early work on the ORA. Overall, I think the methodology is sound. The results are interesting and important. The paper is very terse. I recommend the authors to answer the following questions.

1. Line1: delete "first". I think it's too certain to call this first evidence of hydrothermal reactivation. Hydrothermal reactivation is the "interpretation" for your observation. **Done**
2. Shorten the 'abstract'. Delete first 2 sentences (e.g., in line 10-14) and delete "for instance" in the 3rd sentence. Delete "therefore" in line 16. Delete the sentence around line 21. These changes can make your abstract direct to the point. **Done**
3. Can you provide HPA error bars to fig2a? you mentioned this in line 68-69 but didn't show the error bars.
4. fig2a: If you do 5-day or 10-day averaging of CCT to get HPA? Will that change the HPA results?

If we enlarge the stack window significantly, the HPA measurement would be averaged over the time window of the stack. And if we choose a shorter stack window, the misfit of the rotation ORA would be higher (since the signal to noise ratio of the cross-correlations (CCT) would be lower). Many tests were done to choose and optimise the time window of the stack so that we have the lowest misfit

and the best time resolution. But few days less or more in the stack of the CCT do not change significantly the HPA.

5. Fig3a,b,c: the magnitude of the anisotropy (i.e., length of the red lines) is missing.

The length of the red lines is not the amplitude of anisotropy but the amplitude of the HPA change which is also represented by the colormap.

6. Fig3a,b,c: attach error bars for the Φ_{α} , i.e., a rose diagram of all possible directions.

The error for Φ_{α} is of the order of 2° and is not computed for each direction. This information is added to the text.

7. Line 77: what's the meaning of the word "period" here? It means 'period of time', we changed it in the text.

8. Line 81: in your regionalization method, did you attribute the HPA at the midpoint between the station pair?

The polarization anomaly measurements are considered to be most sensitive to the structure around the two receivers same as the cross-correlation tensor. We use a correlation length of 5 km (around each receiver) in the regionalization.

9. Line87: How does the surface topography around the mountain affect the HPA? As mentioned in the text, the strong topography around at the summit does not allow the fissures to be oriented in the same direction. Hence, the anisotropy is low and the HPA is small (figure 3).

10. Line 110: change "time" to "date". Done

11. Line 115-116: what evidence supports this argument? This is the basic hypothesis of the whole paper: how a change in the fissures, cracks, fluid inclusions in the medium, changes the anisotropy and is an indication of HPA change.

12. Line 118: how much dynamic stress has been changed? Authors should talk about the magnitude of the dynamic stress if this is a reason.

There was no measurement of the dynamic stress in the area of Mount Fuji after the Tohoku earthquake. But an estimation would be around 4MPa.

In Brenguier et al. (2008, Nature Geoscience), they estimated the stress change caused by the velocity perturbation as 2MPa at the Piton de la Fournaise. The velocity change used for this estimation was -1.0×10^{-3} while the maximum average velocity drop in Mount Fuji was -2.0×10^{-3} . Since both of our measurements are held in volcanic area, assuming using the same condition (i.e. the porosity, the incompressibility factor of the medium), we can estimate our dynamic stress change as twice the estimation in the Piton de la Fournaise volcano.

13. Line 126: define "mid-orientation" It is defined in the caption of the figure, but we made it clearer. It means the average orientation between before and after the change at a specific time.

14. Line 388: I think porosity $\phi_0=0.1$ is pretty large, i.e., a 1m cube of rock contains a 0.5m cube fluid? $0.5^3/1^3 \sim 0.1$. is this reasonable?

The minimal porosity for pervasive flows in viscous magmatic systems is usually taken as 0.03 (ref. 29). Using this value will decrease the estimated velocity by a factor 3.3, but will not change the order-of-magnitude- reasoning our conclusion is based on.

Wells in fault systems, that may be closer to the hydrothermal systems considered in the present study, yield porosities between 0.2 to 0.4 (ref. 33). We thus consider 0.1 as a reasonable conservative value.

Reviewer #3:

The authors report on results of changes in seismic anisotropy after the 2011 Tohoku earthquake in the Mt. Fuji area, Japan. Anisotropy is determined by seismic noise cross-correlation. The authors interpret their results as changes in crack alignment that was due to a 'porosity surge' triggered by seismic waves from the Tohoku event.

The study could be of broad interest and is relatively well structured and written. However, I have significant concerns about results and interpretation. The observational evidence for the presented model is, in my opinion, thin and the far-reaching interpretations may not be supported by the data. The authors focus on horizontal polarization anomalies and modest temperature increase, to infer a vertical 'porosity surge' that initiated from a deep magma chamber and strongly changed crack alignments within the affected rock volume. The observations do not unequivocally support this conclusion and no other mechanisms are discussed. A plausible explanation for the observations is simply the mobilization of near surface fluids due to large co-seismic ground motions, resulting in temperature increase. Note that the apparent onset of anisotropy change and temperature increase seem to occur at the same time which would not be expected for a deep source initiating from the magma chamber. Changes in anisotropy may originate from a variety of sources including stress induced anisotropy and fabric related anisotropy. It is difficult to assess the validity of the proposed mechanism in the absence of stronger observational evidence.

As mentioned earlier, we propose a possible interpretation of the results consistent with the geology, geodynamics of the area and the data. That does not mean it is unique, but so far, we did not find any alternative interpretation able to explain so diverse kinds of data.

Indeed seismic anisotropy could have various origins. Extrinsic anisotropy in the subsurface is mainly due to the alignment of fissures and cracks. A **temporal** change of anisotropy can be due to a change in their orientation (induced by a change in the orientation of fissures when subject to a stress perturbation) or a change in its amplitude (induced by the change of the content of the fissures). Our results show a combination of both effects:

1- figure 3 shows a change in the orientation of anisotropy,

2- the strong and non-linear decrease of the HPA (which seems to be dominant) in

figure 2 is interpreted as a change in the amplitude anisotropy due to the fluid surge.

We made this point clearer in the text.

The mobilization of 'near surface fluids' could increase the surface temperature but could not change significantly the anisotropy in the area, unless the fluid is merging from deep depth below the sensitivity area of the measurements.

The modest temperature change as discussed earlier is only one element and as a local measurement it was interesting to mention it. The interpretation is not based on the measurement, but it is consistent with it.

Reviewers' comments:

Reviewer #1 (Remarks to the Author):

The authors answered nearly all points in the reviews, and in most cases provided reasonable explanations and made appropriate changes to the manuscript. In some cases the answers raise further questions, though – I either just use keywords or copy/paste the corresponding points (and only those ones) as this seems to be easiest way to organise the comments in the second review round. I also will comment more generally on the topic of regionalisation and uncertainty. Furthermore I have looked at the Editorial Policy Checklist, and have some further comments on that.

In summary, I believe, the paper is on a good way, but some moderate revisions are still required to clarify some open issues. Even with those, the paper the interpretation put forward in the paper remains a little speculative, but certainly plausible. I am not necessarily convinced but the observation is certainly very interesting and the interpretation is worth putting up for discussion in the scientific literature.

Regionalisation and Uncertainty:

Even after reading the revised version, although it is better described than before, I was still not entirely clear how the regionalisation works. After referring back to Saade et al (2017) I think I finally understand how it works, though. If I am correct, the Tarantola-Valette equation is used with C_m (prior model covariance) defined according to inline equation in l. 384, and the G matrix defined in such a way that the sensitivity is only at the station itself (assuming the inverse problem is set up in terms of a_1 and b_1 it is linear). If the same procedure as in Saade et al. (2017) is followed, then the correlation length λ is chosen by trial and error in a way to minimise the average diagonal element of the posterior covariance matrix (i.e. the value resulting in the maximum reduction from prior to posterior model variance). The supplementary material does not describe how the prior model variance σ_ψ in this equation was chosen. Note that for a correlation length that is too small, only the model in the immediate vicinity of each station will be recovered, leaving the model and model variance at the prior value throughout much of the area. Conversely, for a correlation length that is too large, regions of different anisotropy (change) will be mixed, leading to inconsistent estimates, and thus larger uncertainties. The implication is that the optimal correlation length and variance reduction might depend more strongly on the geometry of the array (e.g., typical distance) and the choice of model prior variance, respectively, than on the uncertainties of the true model. I note that the quoted (model?) variance reduction of 90% is nearly exactly identical to the (model) variance reduction in Saade et al (2017) in a different area, which might be

coincidence but might also hint at that the variance reduction is somehow a property of chosen correlation function.

The upshot of all of this is that

(1) the regionalisation procedure needs to be described more clearly, as the description what is actually inferred (the HPA, as is written) depends on propagation azimuth and is the measurement (data) rather than the model)

(2) the presented result lacks an easily interpretable uncertainty quantification (this applies to many methods in geophysics, so would in my view not exclude publication but needs to be stated clearly not to lead the reader astray). Tarantola and Valette will only provide proper uncertainties if a priori model standard deviation and standard deviations of the data (HPA measurements) are physically based and not adjusted to achieve plausible looking models.

(I have to confess that I am not entirely sure if the above comments apply, as I am not sure I guessed the procedure correctly, but I would say the onus is on the authors to further clarify the description if I got it wrong).

Further comments on rebuttal:

Rev 1 rebuttal (me)

- >> * The period directly after the Tohoku earthquake is characterized by intense
- >> aftershock activity, which might have a strong influence on the quality of the
- >> cross-correlation functions, even when traces are normalized. Can you exclude any
- >> significant effect on the HPA?
- > The aftershocks in the rupture area of the Tohoku-Oki earthquake are smaller than
- > the main earthquake and are quite far, so they cannot have the same influence on
- > the area of Mount Fuji as the more local microseismic activity.

This is not obvious to me. Also the Tohoku aftershocks all come from a similar direction, so would be less likely to be removed from cross-correlations by stacking

- > Besides the
- > aftershock activity in the Tohoku region did not drop suddenly one month after the
- > mainshock.

This, to me is the most convincing argument in your rebuttal of this point.

- > Nonetheless, aftershocks can be considered as additional source of noise and can
- > only increase the signal of the CCTs but they do not change the value of the
- > measured HPA. Especially when the sources of aftershocks are quite far from the
- > network of stations at Mt Fuji and so the source incidence is almost similar at all
- > stations.

I fail to see how this would prevent a bias, and the pattern of apparent HPA change could still be complicated if spurious arrivals interfere with the true HPA.

- > Most importantly, if we look at the cross-correlograms (added figures in the SM), we
- > do not see the effect of the aftershocks on the filtered cross-correlations.

OK

Hakone earthquakes:

- >> However, it is not really well described, where these
- >> earthquakes occur. So I suggest you plot these earthquakes also in map view. It
- >> might get a little busy, but I think you could still fit these into Figure 1, e.g. as small
- >> white circles (change the HiNet stations to squares in this case).

You did not follow or comment on this suggestion – I still don't really know where these earthquakes occur. (Thanks for checking out the possible depth migration – it's a pity this did not show a trend)

- > Concerning the temperature measurement, there are two main effects giving rise to
- > a temperature sudden change: 1- the co-seismic temperature increase due to the
- > perturbation of the fluids close to or on the surface and the start of the purge from
- > the deeper hydrothermal zone, 2- the complete purge of the system and the arrival
- > at the surface of all the fluids that fill the cracks in the medium.
- > Note that, the return to normal of the surface temperature is slower than that of
- > anisotropy, since the temperature diffusion is slower.

These two effects seem to be more or less independent, it is remarkable that this then leads to the boxcar shaped anomaly (in time) with a quick reversal of the anomaly back to the background state after one month. I guess we have to accept that this is something not yet fully understood.

Also, is temperature diffusion even relevant, as I imagine temperature transport is accomplished mostly by advection?

>> All: Why don't you show the data for the whole time you processed, i.e. starting

>> earlier. The signal is quite suggestive, but there are also some (apparent?) HPA

>> changes almost reaching the 2-sigma level in February, and it left me wondering

>> how frequent such excursions are, and how significant the one observed after the

>> Tohoku earthquake actually is.

> The fluctuations (for instance, in February) can be related to local effects, such as

> seasonal changes. But the important observation is that the co-seismic change is

> quite spectacular and occurs at the time of the Tohoku-Oki earthquake.

The anomaly in February is too short-lived to be described as seasonal. Of course it could be related to weather or hydrological events like rainfall. I still think it is a pity that not a longer time series has been processed. The argument that the main anomaly appears just at the day of the Tohoku earthquake is quite persuasive, but still it would still be interesting if anomalies of similar size also occur independently of the Tohoku earthquake (of course, you could argue this is beyond the scope of the article).

On plotting also the static values in Fig 3

> We only have less than 2 months of measurements before the Tohoku-Oki

> earthquake, where we see a lot of fluctuations in the HPA measurements. A

> regionalisation of the static anisotropy in the period would not be reliable.

Are you saying that a 2-month average is not stable but the change detection over 7 days is? I find it hard to reconcile these two statements

Fig. 3

>> - (a)-(c) Is the legend for the color scale mislabeled. It looks like the color is chosen

>> according to the anisotropy amplitude, i.e. α (and not the average fast direction
>> as is indicated, i.e. Ψ_a).

> The color bar represents the values of the HPA. Thank

> you, we corrected the labels in the figures.

I don't understand how plotting the HPA is sensible, see comments below.

Rev 2 rebuttal:

>> 6. Fig3a,b,c: attach error bars for the Φ_α , i.e., a rose diagram of all possible
>> directions.

> The error for Φ_α is of the order of 2° and is not computed for each direction.

> This information is added to the text.

How do you determine this error?

>> 12. Line 118: how much dynamic stress has been changed? Authors should talk
>> about the magnitude of the dynamic stress if this is a reason.

> There was no measurement of the dynamic stress in the area of Mount Fuji after
> the Tohoku earthquake. But an estimation would be around 4MPa.

> In Brenguier et al. (2008, Nature Geoscience), they estimated the stress change

> caused by the velocity perturbation as 2MPa at the Piton de la Fournaise. The

> velocity change used for this estimation was -1.0×10^{-3} while the maximum

> average velocity drop in Mount Fuji was -2.0×10^{-3} . Since both of our

> measurements are held in volcanic area, assuming using the same condition (i.e.

> the porosity, the incompressibility factor of the medium), we can estimate our

> dynamic stress change as twice the estimation in the Piton de la Fournaise volcano.

Dynamic stress normally refers to the stress perturbation as the wave is passing through (and your reference to surface waves shows you are using it in the same way). The estimate of Brenguier seems to refer to the static stress (I guess Florent is a co-author and so should know better than me, but the description makes sense only in the context of static stress). The dynamic stress could be

estimated from synthetic seismogram calculation for Tohoku-Oki earthquake (I personally wouldn't think it's that helpful to know the stress perturbation, though).

Comments on editorial policy statement:

Data availability:

The authors write "The data sets analyzed during the current study are available from the corresponding author on reasonable request"

I would not consider this a full data availability statement. The original seismic data used come from different networks, at least some of which are actually open data (e.g., HiNet data). The statement should either give data DOIs, FDSN network codes, or web addresses where the original waveform data can be accessed, and if they are open or restricted. I would also strongly encourage to include relevant data sets as supplementary information or archive them in a repository, e.g. institutional, as 'available on request from the author' can fail in many ways (author left science, retired, is away on a long field trip, accidentally lost harddrive with the data etc).

"All relevant accession codes are provided" was ticked – I did not see any accession codes in the material provided to me (to be honest I don't know exactly what they are but I presume some formalised links to the data, such as data DOIs used in seismology).

"We have provided a full code availability statement in the manuscript" - I did not see this statement in the manuscript. Only reference to code is in the author contribution statement, where it is written which author contributed which code. But it is not clear whether and how these codes are accessible to scientists outside the working group.

Data presentation

No box plots are shown so the question on 'box plot elements' should be n/a

"Clearly defined error bars are present and what they present is noted" - This is true for the error shown in Extended Data Fig. 4, but in Fig. 2 the bar is just explained as showing the "error" but it is not clear what this error represents.

Comments on the text and figures:

The revisions seemed to have introduced a rather large number of minor English language issues at a rate higher than in the original submission, so authors should carefully double-check the final version before submission.

p 3 | 37 "... OTHER volcanic areas WORLDWIDE" – to make clear you are not talking about Japan and the response to Tohoku-Oki earthquake anymore.

p 3 | 40 crack → cracks, delete "distribution in a seismogenic zone" – note that the term seismogenic zone is usually associated with the seismicity generating parts of major faults. I think here you want to express "in a seismically active region" but this is already implied by the fact that it is a coseismic change.

p 3 | 55 omitted in the medium → neglected

p 3 | 57 also for → can also be affected by

p 3 | 59-61 seasonal variations or changes in seismic anisotropy: This is not a dichotomy. The seasonal variations might actually be caused by seasonal changes in seismic anisotropy (e.g. due to rainfall induced changes of crack-induced anisotropy). Alternatively, the seasonal variation of noise source distributions could cause spurious changes in HPA. A third possibility would be that the heterogeneous structure changes and causes variation in HPA. Which "seasonal variation" do you refer to here, please clarify the text. Also, you state that the seasonal variation can be separated, but the processing does not seem to involve any removal of such a component (which is fine). If so, Fig. 2 and Extended Fig. 3, 4 seem to hint at a weak seasonal dependence only (substantially smaller than the anomaly of interest). Based on one year of data I don't think it is reasonable to remove the seasonal signal, as the estimate would be skewed by the anomalies.

p 3 | 69 spreads → appears to spread

p 3 | 76 during THE TIME period [] ..

p 5 | 133 crack-filled water → cracks filled with water

p 5 | 162 fluids that filled the cracks

Fig. 2 caption Add degree sign ($^{\circ}$) after 2 in the statement on errors. How do you know this is the error, though? It seems it could be based on the observed standard deviation, but you have no idea if this is a measurement error, or uninterpreted signal (i.e. related to true changes of the anisotropy) – particularly the standard deviation estimate will be influenced significantly by the weak seasonal (or otherwise long period) variation. An easy solution would be to remove error, and write standard deviation or 2σ (but then give the actual value, and not ‘order of 2’).

Also, how is the averaging of the HPA done, given that its actual value depends on the propagation azimuth? (see my comment on Fig 3 for more details)

Fig. 3 I am still confused what exactly is plotted here. The caption and legend claim it is HPA, which I understand to be ψ_P as defined in eq. 1 and 2 in the supplement. But this is highly dependent on the propagation azimuth, so does not really make sense to regionalise HPA per se. What would make sense – to me at least – is to regionalise the values of α and ψ_a (or more easily a_1 and b_1 , and then plot the values of α and ψ_a derived from the local (a_1, b_1) pair).

p 14 | 363 “atan” should not be in Italics

p 14 | 370 smoothened → smoothed

p 14 | 375 deriving the equation of ψ_P → taking the derivative of equation (2)

In eq 3, finite differences are approximated with infinitesimal differences, i.e. the equation will only be valid if $\Delta \psi_{\alpha} \ll 180^{\circ}$ and $\Delta \alpha \ll \alpha$. Is this the case?

p 14 | 381 The polarisation anomaly depends on the azimuth of propagation, i.e., approximately the interstation azimuth (eq 2) but you estimate this property just as a function of spatial position. Are you sure you do not regionalise over α or ψ_{α} instead (see comment on Fig 3)?

p 14 | 386 wavelength → correlation length (I think – the wavelength would probably be a reasonable correlation length)

p 14 | 387 resulted → resultant , can explain THE data

p 14 | 388 Please clarify with what you mean by variance reduction here; normally variance reduction is used to refer to the reduction of residuals from the initial or reference model to the preferred model. But I have a feeling you use might it to describe the difference between prior and posterior model variance.

p 16 | 432 Delete “reasonable”

Reviewer #3 (Remarks to the Author):

The revised manuscript by Saade et al. includes several improvements to the initial submission, including some more details about the applied methods. Unfortunately, the authors missed one of my main comments: “Note that the apparent onset of anisotropy change and temperature increase seem to occur at the same time which would not be expected for a deep source initiating from the magma chamber. ”, which was also echoed by reviewer 1:

“According to the model in Fig 4, the porosity is increased initially at depth and only then propagates upward. So why would I expect an immediate temperature increase after the Tohoku EQ?”

The comment is related to my general concern that the presented observations are quite limited and the author’s interpretation of the observations seems implausible based on the temperature data.

The additional information provided in the rebuttal letter now shows aftershocks that start shallow and become deeper with time after Tohoku (so opposite of what is expected based on their model). They also admit that their method to resolve velocity differences is insensitive to depth (from the rebuttal):

“Unfortunately we don’t have access to more data from more seismic stations and the procedure of getting the data converting and processing it is quite long and beyond the scope of this paper Indeed the frequency range should be chosen precisely it should cover one micro-seismic peak and give good quality cross-correlations from which we have a minimized misfit at the rotation. The selected frequency range is relatively narrow so we have no depth resolution. “

The authors mention that they have no depth resolution and the measurement is averaged over 7 days. Given the above caveats and lack of other supporting information, an interpretation of the origin of the anisotropy seems very problematic, even if the measurements of anisotropy changes are robust. Publishing the article without a clearer understanding of underlying mechanisms seems to have limited value.

Dear Editor and reviewers,

We would like to thank you again for evaluating our manuscript entitled “Evidence of reactivation of a hydrothermal system from seismic anisotropy changes”. We are grateful for your comments and we believe they improved the manuscript.

We made significant changes mainly to the text and we tried to follow your suggestions. The English language was corrected by a professional editor.

For instance, we further detailed the regionalisation technique in the supplementary methods. We also provided links to download the data.

The third reviewer’s concern is mainly related to the explanation of two temperature increases. We tried to clarify that two different processes are associated with the two increases of temperature: The initial coseismic increase of surface temperature in the springs is consistent with the permeability increase induced by the realignment of cracks as a consequence of coseismic stress perturbation induced by the Tohoku-Oki earthquake. The second extra increase of surface temperature, almost one month after the earthquake, is due to the arrival of the porous surge and hot fluids to the surface from the 3km deep hydrothermal reservoir.

We added further arguments and explanation on the link between anisotropy and temperature change, in the response to the reviewer and in the text.

We hope that this new revised version of the manuscript answers all your questions and concerns and is suitable for publication in Nature Communication.

Looking forward to hearing from you.

Best regards,

On behalf of the authors,
Maria Saadé

Below are point by point response to the reviewers' comments:

Reviewer #1 (Remarks to the Author):

Regionalisation and Uncertainty:

Even after reading the revised version, although it is better described than before, I was still not entirely clear how the regionalisation works. After referring back to Saade et al (2017) I think I finally understand how it works, though. If I am correct, the Tarantola-Valette equation is used with C_m (prior model covariance) defined according to inline equation in l. 384, and the G matrix defined in such a way that the sensitivity is only at the station itself (assuming the inverse problem is set up in terms of a_1 and b_1 it is linear). If the same procedure as in Saade et al. (2017) is followed, then the correlation length λ is chosen by trial and error in a way to minimise the average diagonal element of the posterior covariance matrix (i.e. the value resulting in the maximum reduction from prior to posterior model variance).

The supplementary material does not describe how the prior model variance σ_ψ in this equation was chosen.

MS: Yes, you are right, the regionalisation is extensively detailed in Saade et al. (GJI, 2017) based on the data covariance matrix (diagonal with error bars) and the model covariance function characterized by prior model error and a correlation length. The same data error is taken for all pairs of stations, equal to 0.035rad (2degrees) and the a priori error on σ_ψ is 25degrees. This choice is somehow arbitrary but dictated by what was observed in other tectonic contexts. It affects the amplitude of the inverted ψ (and the posterior model errors) but not their lateral variations. In other words, whatever σ_ψ , the regions of large or small ψ_P are at the same locations but with a slightly larger or smaller amplitude.

Note that for a correlation length that is too small, only the model in the immediate vicinity of each station will be recovered, leaving the model and model variance at the prior value throughout much of the area. Conversely, for a correlation length that is too large, regions of different anisotropy (change) will be mixed, leading to inconsistent estimates, and thus larger uncertainties. The implication is that the optimal correlation length and variance reduction might depend more strongly on the geometry of the array (e.g., typical distance) and the choice of model prior variance, respectively, than on the uncertainties of the true model. I note that the quoted (model?) variance reduction of 90% is nearly exactly identical to the (model) variance reduction in Saade et al (2017) in a different area, which might be coincidence but might also hint at that the variance reduction is somehow a property of chosen correlation function.

The upshot of all of this is that

(1) the regionalisation procedure needs to be described more clearly, as the description what is actually inferred (the HPA, as is written) depends on propagation azimuth and is the measurement (data) rather than the model)

(2) the presented result lacks an easily interpretable uncertainty quantification (this applies to many methods in geophysics, so would in my view not exclude publication but needs to be stated clearly not to lead the reader astray). Tarantola and Valette will only provide proper uncertainties if a priori model standard deviation and standard deviations of the data (HPA measurements) are physically based and not adjusted to achieve plausible looking models. (I have to confess that I am not entirely sure if the above comments apply, as I am not sure I guessed the procedure correctly, but I would say the onus is on the authors to further clarify the description if I got it wrong).

MS: Please see the section “Anisotropy Regionalization Method” in the supplementary methods in which we further detailed the regionalisation method.

Further comments on rebuttal:

Rev 1 rebuttal (me)

>> * The period directly after the Tohoku earthquake is characterized by intense
>> aftershock activity, which might have a strong influence on the quality of the
>> cross-correlation functions, even when traces are normalized. Can you exclude any
>> significant effect on the HPA?

> The aftershocks in the rupture area of the Tohoku-Oki earthquake are smaller than
> the main earthquake and are quite far, so they cannot have the same influence on
> the area of Mount Fuji as the more local microseismic activity.

*This is not obvious to me. Also, the Tohoku aftershocks all come from a similar direction, so
would be less likely to be removed from cross-correlations by stacking*

> Besides the

> aftershock activity in the Tohoku region did not drop suddenly one month after the
> mainshock.

This, to me, is the most convincing argument in your rebuttal of this point.

> Nonetheless, aftershocks can be considered as additional source of noise and can

> only increase the signal of the CCTs but they do not change the value of the

> measured HPA. Especially when the sources of aftershocks are quite far from the

> network of stations at Mt Fuji and so the source incidence is almost similar at all

> stations.

*I fail to see how this would prevent a bias, and the pattern of apparent HPA change could still
be complicated if spurious arrivals interfere with the true HPA.*

**MS: The sources of microseismic noise are primarily local (Japan, Philippines seas and
Pacific ocean). The aftershocks of Tohoku earthquake (several hundreds of kilometers away)
amplify the noise amplitude and so the signal on the CCTs. If the source of noise, here the
aftershocks, is omnidirectional or unidirectional and aligned with the station pair, the HPA
(which is related to anisotropy) does not change. It is only the signal on the CCTs that
changes. But again the aftershocks are not visible on the filtered CCTs.**

> Most importantly, if we look at the cross-correlograms (added figures in the SM), we

> do not see the effect of the aftershocks on the filtered cross-correlations.

OK

Hakone earthquakes:

>> However, it is not really well described, where these

>> earthquakes occur. So, I suggest you plot these earthquakes also in map view. It

>> might get a little busy, but I think you could still fit these into Figure 1, e.g. as small

>> white circles (change the HiNet stations to squares in this case).

*You did not follow or comment on this suggestion – I still don't really know where these
earthquakes occur. (Thanks for checking out the possible depth migration – it's a pity this did
not show a trend)*

MS: We added the map with the aftershocks location in the Extended data section.

> Concerning the temperature measurement, there are two main effects giving rise to

> a temperature sudden change: 1- the co-seismic temperature increase due to the

> perturbation of the fluids close to or on the surface and the start of the purge from

> the deeper hydrothermal zone, 2- the complete purge of the system and the arrival

> at the surface of all the fluids that fill the cracks in the medium.

> Note that, the return to normal of the surface temperature is slower than that of

> anisotropy, since the temperature diffusion is slower.

*These two effects seem to be more or less independent, it is remarkable that this then leads
to the boxcar shaped anomaly (in time) with a quick reversal of the anomaly back to the
background state after one month. I guess we have to accept that this is something not yet
fully understood.*

Also, is temperature diffusion even relevant, as I imagine temperature transport is

accomplished mostly by advection?

MS: Yes, the temperature changes following the Tohoku earthquake and one month later are quite puzzling (as also mentioned by reviewer #3). Both advection and diffusion could occur during the temperature increase and decrease. Thermal advection is a non-linear and relatively rapid phenomena that could be related to the return to the depth of the hydrothermal fluids. And thermal diffusion is a slower phenomena that could be due to the cooling of surface fluid/water in the dam.

>> *All: Why don't you show the data for the whole time you processed, i.e. starting earlier. The signal is quite suggestive, but there are also some (apparent?) HPA changes almost reaching the 2-sigma level in February, and it left me wondering how frequent such excursions are, and how significant the one observed after the Tohoku earthquake actually is.*

> *The fluctuations (for instance, in February) can be related to local effects, such as seasonal changes. But the important observation is that the co-seismic change is quite spectacular and occurs at the time of the Tohoku-Oki earthquake.*

The anomaly in February is too short-lived to be described as seasonal. Of course it could be related to weather or hydrological events like rainfall. I still think it is a pity that not a longer time series has been processed. The argument that the main anomaly appears just at the day of the Tohoku earthquake is quite persuasive, but still it would still be interesting if anomalies of similar size also occur independently of the Tohoku earthquake (of course, you could argue this is beyond the scope of the article).

MS: Of course it would have been much more interesting to process many years of data, but unfortunately we do not own the data and don't have access to more data.

But anyway, the strong changes that we interpret, not only occurred at the time of the Tohoku earthquake and one month later but are by far the fastest changes during the year 2011 that we processed (see Figure Extended data 1 in SM).

On plotting also the static values in Fig 3

> *We only have less than 2 months of measurements before the Tohoku-Oki earthquake, where we see a lot of fluctuations in the HPA measurements. A regionalisation of the static anisotropy in the period would not be reliable.*

Are you saying that a 2-month average is not stable but the change detection over 7 days is? I find it hard to reconcile these two statements

MS: Regionalizing the average static HPA or anisotropy is not the same as regionalizing their temporal changes (see equations 2 and 3 in SM). Besides, the temporal change of HPA is the difference between daily HPAs computed each from CCTs stacked over 7 days. The static HPA is computed from the CCTs stacked over a large time window, here the whole year. The shorter is this time window the more uncertainty we have on the static HPA, hence the static anisotropy in the medium.

Fig. 3

>> *-(a)-(c) Is the legend for the color scale mislabeled. It looks like the color is chosen according to the anisotropy amplitude, i.e. α (and not the average fast direction as is indicated, i.e. Ψ_a).*

> *The color bar represents the values of the HPA. Thank you, we corrected the labels in the figures.*

I don't understand how plotting the HPA is sensible, see comments below.

Rev 2 rebuttal:

>> 6. Fig3a,b,c: attach error bars for the Φ_{α} , i.e., a rose diagram of all possible >> directions.

> The error for Φ_{α} is of the order of 2° and is not computed for each direction.

> This information is added to the text.

How do you determine this error?

MS: The error on the fast HPA change is considered to be of the order of the HPA continuous fluctuations during calm periods (i.e. period where no event or fast change is observed). From figure 2a we see that these small fluctuations (i.e. between May and end of 2011) reach maximum 2° . This point is clarified in the text.

>> 12. Line 118: how much dynamic stress has been changed? Authors should talk >> about the magnitude of the dynamic stress if this is a reason.

> There was no measurement of the dynamic stress in the area of Mount Fuji after > the Tohoku earthquake. But an estimation would be around 4MPa.

> In Brenguier et al. (2008, Nature Geoscience), they estimated the stress change

> caused by the velocity perturbation as 2MPa at the Piton de la Fournaise. The

> velocity change used for this estimation was -1.0×10^{-3} while the maximum

> average velocity drop in Mount Fuji was -2.0×10^{-3} . Since both of our

> measurements are held in volcanic area, assuming using the same condition (i.e.

> the porosity, the incompressibility factor of the medium), we can estimate our

> dynamic stress change as twice the estimation in the Piton de la Fournaise volcano.

Dynamic stress normally refers to the stress perturbation as the wave is passing through

(and your reference to surface waves shows you are using it in the same way). The estimate

of Brenguier seems to refer to the static stress (I guess Florent is a co-author and so should know better than me, but the description makes sense only in the context of static stress).

The dynamic stress could be estimated from synthetic seismogram calculation for

Tohoku-Oki earthquake (I personally wouldn't think it's that helpful to know the stress perturbation, though).

MS: Sorry about this confusion. The dynamic stress was discussed in Brenguier et al.(2014). In the Mount Fuji area it is estimated to be of the order of 1MPa.

Comments on editorial policy statement:

Data availability:

The authors write "The data sets analyzed during the current study are available from the corresponding author on reasonable request"

I would not consider this a full data availability statement. The original seismic data used come from different networks, at least some of which are actually open data (e.g., HiNet data). The statement should either give data DOIs, FDSN network codes, or web addresses where the original waveform data can be accessed, and if they are open or restricted. I would also strongly encourage to include relevant data sets as supplementary information or archive them in a repository, e.g. institutional, as 'available on request from the author' can fail in many ways (author left science, retired, is away on a long field trip, accidentally lost

harddrive with the data etc).

“All relevant accession codes are provided” was ticked – I did not see any accession codes in the material provided to me (to be honest I don’t know exactly what they are but I presume some formalised links to the data, such as data DOIs used in seismology).

“We have provided a full code availability statement in the manuscript” - I did not see this statement in the manuscript. Only reference to code is in the author contribution statement, where it is written which author contributed which code. But it is not clear whether and how these codes are accessible to scientists outside the working group.

We probably misunderstood the statements about the availability of the data and will clarify it in the revised MS. We provided the link to download the seismic data:

<http://www.bosai.go.jp/e/activities/database/>

Data presentation

No box plots are shown so the question on ‘box plot elements’ should be n/a

“Clearly defined error bars are present and what they present is noted” - This is true for the error shown in Extended Data Fig. 4, but in Fig. 2 the bar is just explained as showing the “error” but it is not clear what this error represents.

Point clarified previously in this rebuttal letter.

Comments on the text and figures:

The revisions seemed to have introduced a rather large number of minor English language issues at a rate higher than in the original submission, so authors should carefully double-check the final version before submission.

p 3 | 37 “... OTHER volcanic areas WORLDWIDE” – to make clear you are not talking about Japan and the response to Tohoku-Oki earthquake anymore.

Corrected

p 3 | 40 crack → cracks, delete “distribution in a seismogenic zone” – note that the term seismogenic zone is usually associated with the seismicity generating parts of major faults. I think here you want to express “in a seismically active region” but this is already implied by the fact that it is a coseismic change.

Corrected

p 3 | 55 omitted in the medium → neglected

Corrected

p 3 | 57 also for → can also be affected by

Corrected

p 3 | 59-61 seasonal variations or changes in seismic anisotropy: This is not a dichotomy. The seasonal variations might actually be caused by seasonal changes in seismic anisotropy (e.g. due to rainfall induced changes of crack-induced anisotropy). Alternatively, the seasonal

variation of noise source distributions could cause spurious changes in HPA. A third possibility would be that the heterogeneous structure changes and causes variation in HPA. Which “seasonal variation” do you refer to here, please clarify the text. Also, you state that the seasonal variation can be separated, but the processing does not seem to involve any removal of such a component (which is fine). If so, Fig. 2 and Extended Fig. 3, 4 seem to hint at a weak seasonal dependence only (substantially smaller than the anomaly of interest). Based on one year of data I don't think it is reasonable to remove the seasonal signal, as the estimate would be skewed by the anomalies.

MS: Indeed anisotropy change can also be seasonal. We do agree with your comment, it was probably not very clear in the text (we made changes to the text). The idea was to separate between slow seasonal changes and rapid co-seismic changes (here seismic anisotropy) that we observe on the HPA continuous measurements.

In some cases seasonal changes can be large and can hide co-seismic changes. So, seasonal and co-seismic changes can be separated by applying a singular value decomposition filter as in the case of Saade et al. 2017. In the case of this study, observed seasonal variations are weak and co-seismic changes are quite visible and so the separation is not needed and is not reasonable as you say.

p 3 | 69 spreads → appears to spread

Corrected

p 3 | 76 during THE TIME period [] ..

Corrected

p 5 | 133 crack-filled water → cracks filled with water

Corrected

p 5 | 162 fluids that filled the cracks

Corrected

Fig. 2 caption Add degree sign (°) after 2 in the statement on errors. How do you know this is the error, though? It seems it could be based on the observed standard deviation, but you have no idea if this is a measurement error, or uninterpreted signal (i.e. related to true changes of the anisotropy) – particularly the standard deviation estimate will be influenced significantly by the weak seasonal (or otherwise long period) variation. An easy solution would be to remove error, and write standard deviation or 2σ (but then give the actual value, and not ‘order of 2’).

MS: The estimation of the error is discussed previously in this rebuttal letter: “The error on the fast HPA change is considered to be of the order of the HPA continuous fluctuations during calm periods (i.e. period where no event or fast change is observed). From figure 2.a we see that these small fluctuations (i.e. between May and end of 2011) reach a maximum of 2°. This point was clarified in the text.”

Also, how is the averaging of the HPA done, given that its actual value depends on the propagation azimuth? (see my comment on Fig 3 for more details)

Fig. 3 I am still confused what exactly is plotted here. The caption and legend claim it is HPA, which I understand to be ψ_P as defined in eq. 1 and 2 in the supplement. But this is highly dependent on the propagation azimuth, so does not really make sense to regionalise HPA per se. What would make sense – to me at least – is to regionalise the values of α and ψ_a (or more easily a_1 and b_1 , and then plot the values of α and ψ_a derived from the local (a_1, b_1) pair).

MS: The technique presented in the paper does not allow to invert the amplitude of anisotropy (α). We compute the HPA and invert the orientation of anisotropy (Ψ_{α}). And we regionalise both the HPA and Ψ_{α} . The orientation of the red lines represent the orientation of seismic anisotropy and the color code represents the amplitude of the HPA. It is true that the HPA depends on the propagation azimuth. In our case the noise sources are far enough from the seismic network. Consequently the noise incidence is almost the same at all stations and the HPA are almost equal at both stations of a pair. This helps remove the contribution of the noise source azimuth.

Concerning the azimuth of the station pair: the HPA is the polarization anomaly from which the azimuth of the station is derived. HPA = Polarization of quasi-surface waves - azimuth of the station pair.

p 14 | 363 "atan" should not be in Italics

Corrected

p 14 | 370 smoothened → smoothed

Corrected

p 14 | 375 deriving the equation of ψ_P → taking the derivative of equation (2)

Corrected

In eq 3, finite differences are approximated with infinitesimal differences, i.e. the equation will only be valid if $\Delta \psi_{\alpha} \ll 180^{\circ}$ and $\Delta \alpha \ll \alpha$. Is this the case?

MS: It is. The horizontal polarization anomaly is computed every hour using a correlation tensor stacked over 7 days (centered by this hour). Changes in the orientation or amplitude of anisotropy that can occur in this range of time are most probably smaller than respectively 180° and Δ_{α} .

p 14 | 381 The polarisation anomaly depends on the azimuth of propagation, i.e., approximately the interstation azimuth (eq 2) but you estimate this property just as a function of spatial position. Are you sure you do not regionalise over α or ψ_{α} instead (see comment on Fig 3)?

MS: point clarified previously.

p 14 | 386 wavelength → correlation length (I think – the wavelength would probably be a reasonable correlation length)

Corrected

p 14 | 387 resulted → resultant, can explain THE data

Corrected

p 14 | 388 Please clarify with what you mean by variance reduction here; normally variance reduction is used to refer to the reduction of residuals from the initial or reference model to the preferred model. But I have a feeling you use might it to describe the difference between prior and posterior model variance.

MS: Here the variance describe our a priori knowledge on parameters such as the correlation length. So we try to reduce the a posteriori variance relative to the a priori variance. This info is added to the text.

p 16 | 432 Delete "reasonable"

Corrected

Reviewer #3 (Remarks to the Author):

The revised manuscript by Saade et al. includes several improvements to the initial submission, including some more details about the applied methods. Unfortunately, the authors missed one of my main comments: “Note that the apparent onset of anisotropy change and temperature increase seem to occur at the same time which would not be expected for a deep source initiating from the magma chamber. ”, which was also echoed by reviewer 1:

“According to the model in Fig 4, the porosity is increased initially at depth and only then propagates upward. So why would I expect an immediate temperature increase after the Tohoku EQ?”

MS: Yes, you are right, we were also puzzled by this increase in temperature following the Tohoku earthquake, and as already explained to the reply to reviewer #1, we propose that two processes can explain these two changes. We consider that the change of seismic anisotropy induced by the Tohoku-Oki earthquakes is related to changes in the alignment of cracks that in turn open the permeability of the hydrothermal system. The initial increase of temperature of the hot spring waters (of $\approx 3^{\circ}\text{C}$) associated with the HPA change is consistent with such a permeability increase and a first shallow degassing. If the flow in the porous medium was a homogeneous Darcy's flow, one would expect the thermal anomaly measured in the springs to be maintained as long as the permeability remains open. However, because of the non-linear relationship between porosity and permeability in porous systems, a homogeneous Darcy's flow is not stable and rather bounds to produce porosity surges, possibly solitary waves or porosity shocks as a function of the effective rheology of the medium [e.g. 33, 34]. When such a surge originating approximately at a depth of 3km, reaches the surface it shall produce an extra increase of the temperature in the springs (here an additional 2°C thermal anomaly) because the flux of hot fluids increases. We furthermore show in the paper that the velocity of a porosity surge is consistent with the one month delay between the activation of the hydrothermal system and the thermal peak recorded in the springs. Last, because the quantity of fluid is much larger in the surge than through the background flow we consider that the surge extract enough fluid from the system to go back to its initial stage hence for return to the original seismic anisotropy state.

Further clarification of the relation between seismic anisotropy and temperature change is added to the text.

The comment is related to my general concern that the presented observations are quite limited and the author's interpretation of the observations seems implausible based on the temperature data.

MS: we expect that this additional discussion will convince the reviewer that the two temperature increases are not contradictory. In addition, the most important observation and the central part of our results regards the rapid change in polarization anomalies and we believe that seismicity and temperature changes add some consistent pieces to our interpretation

The additional information provided in the rebuttal letter now shows aftershocks that start shallow and become deeper with time after Tohoku (so opposite of what is expected based on their model).

MS: The figure of aftershock migration added in the previous rebuttal letter shows that there is no clear depth migration.

They also admit that their method to resolve velocity differences is insensitive to depth (from

the rebuttal):

“Unfortunately we don’t have access to more data from more seismic stations and the procedure of getting the data converting and processing it is quite long and beyond the scope of this paper. Indeed the frequency range should be chosen precisely it should cover one micro-seismic peak and give good quality cross-correlations from which we have a minimized misfit at the rotation. The selected frequency range is relatively narrow so we have no depth resolution. “

The authors mention that they have no depth resolution and the measurement is averaged over 7 days. Given the above caveats and lack of other supporting information, an interpretation of the origin of the anisotropy seems very problematic, even if the measurements of anisotropy changes are robust. Publishing the article without a clearer understanding of underlying mechanisms seems to have limited value.

MS: Until today temporal resolution of monitored physical parameters that are computed using passive interferometry (ambient noise cross-correlation) is of the order of several days. We need to stack the cross-correlations over several days in order to improve the signal to noise ratio.

The method we present in this paper can be improved, but it will be difficult to escape the 7 days stack due to the microseismic sources.

We believe that we propose a reasonable mechanism of surge and it is the first time that this kind of phenomenon is observed. There is still a lot to explore. But we think that the observations we have and their possible interpretations are quite interesting and promising. This paper should be the basis of the further exploration of this method for passive monitoring of the crust using seismic anisotropy.

Reviewers' comments:

Reviewer #1 (Remarks to the Author):

Thank you for taking the time to go thoroughly through all the reviewer comments and addressing most of them. Overall, the paper is in quite good shape now. In particular, there is now a description of how the uncertainty of measurements was estimated. While the interpretation remains a tad speculative, it is certainly a plausible model, which can explain the anisotropy change observations. However, I am still a bit mystified by the explanation in the supplementary material (and partially in the rebuttal letter) on the regionalisation of the static anisotropy and its derivative, and what is being solved for. I am almost certain that this lack of understanding is only pointing to the clarity of explanation rather than a deeper issue, and blame might also be on my side for not being able to follow your description.

I will try to express my difficulties based on an example. The way I understand it, HPA is Ψ_P in eq 1, 2, and "orientation of the 2 Ψ anisotropy" is Ψ_α (it would be helpful to the reader if this were explicitly stated; and then the symbols used preferentially at least in the supplementary material).

Assuming a model with uniform anisotropy with fast north direction, and four stations arranged in a square:

A B ^

|

D C

according to equation 2,

$\Psi_P=0$ for paths A-B, D-C, A-D and B-C

$\Psi_P=-\alpha$ for D-B

and $\Psi_P=+\alpha$ for A-C

You write (l 393-394) "For every grid point of 2.2 km x 2.2 km, the average of the horizontal polarization anomalies is computed ...". However, the average of the values for Ψ_P over the different paths would be 0 for the example above but this is misleading with regard to the true anisotropy; the same averaging to zero or nearly zero would also occur for very dense azimuthal coverage. Taking an average of random samples of an oscillation is generally not very informative.

Further you write in the rebuttal letter "The technique presented in the paper does not allow to invert the amplitude of anisotropy (α)" but with the observations of Ψ_P at different azimuths you would have sufficient information to recover both α and Ψ_α . In a uniform model this would simply be a matter of fitting the phase and amplitude of a sine-wave based on a few

Ψ_P measurements at different azimuths. There is no way to meaningfully estimate Ψ_α without also estimating α except in very contrived circumstances (if all the azimuths you had were very close to zero-crossing I guess you could have very low resolution of α and good resolution of Ψ_α)

Regionalisation is fundamentally a way of spatial averaging. As even the sign of Ψ_P depends on the propagation azimuth, and by necessity the regionalisation mixes paths with different azimuths, I don't understand what the regionalisation would represent. I could speculate that maybe you use the absolute value of Ψ_P , i.e. $|\Psi_P|$ for the regionalisation. This would be borderline OK, as on average this value would be proportional to α , but still you would get a bias depending on the actual mix of contributing azimuths. Solving for regionalised α and Ψ_α should still give you superior results in this situation, too.

Similar reasoning can be applied to equation 3 on the changes. $\Delta \Psi_P$ can be positive or negative depending on the azimuth of propagation, and if measurements from a few azimuths are available it should be possible to deduce both $\Delta \alpha$ and $\Delta \Psi_\alpha$.

Further comments on the rebuttal letter:

p2

- > ... The same data error is
- > taken for all pairs of stations, equal to 0.035rad (2degrees) and the a priori error on
- > σ_Ψ is 25degrees. This choice is somehow arbitrary but dictated by what was
- > observed in other tectonic contexts. It affects the amplitude of the inverted Ψ (and the
- > posterior model errors) but not their lateral variations. In other words, whatever σ_Ψ ,
- > the regions of large or small Ψ_P are at the same locations but with a slightly larger or
- > smaller amplitude.

Thank you, this is helpful. I suggest to report this information (including the values used for a priori error in the supplementary material).

p 6 of rebuttal letter

>> "We have provided a full code availability statement in the manuscript" - I did not see this

>> statement in the manuscript. Only reference to code is in the author contribution statement,
>> where it is written which author contributed which code. But it is not clear whether and how
>> these codes are accessible to scientists outside the working group.

> We probably misunderstood the statements about the availability of the data and will clarify it

> in the revised MS. We provided the link to download the seismic data:

> <http://www.bosai.go.jp/e/activities/database/>

\thank you for providing the link on seismic data availability (very good!), but what about “code availability”.

P 9 of rebuttal letter

> We consider that the change of seismic

> anisotropy induced by the Tohoku-Oki earthquakes is related to changes in the alignment of

> cracks that in turn open the permeability of the hydrothermal system. The initial increase of

> temperature of the hot spring waters (of $\approx 3^{\circ}\text{C}$) associated with the HPA change is consistent

> with such a permeability increase and a first shallow degassing.

In addition to a possible coseismic permeability increase, another effect is shaking induced compaction of porous layers, which then reduces porosity, in turn increasing hydraulic pressure. This is the effect behind liquefaction, but in a milder form will also lead to upward surges of fluids.

Further comments regarding manuscript:

Main text

p 6 273 Add degree sign to temperature, i.e. 2°C

Suppl Material:

In the whole added section, note that units like “km” should not be italicised.

P 15 l 393 grid point \rightarrow grid cell

l 399 “an empirical correlation length” - add which value you actually used in km (I guess 5 km ?)

p 16 | 415-417 As written, the sentence does not make much sense. Did you mean to say “that this choice of correlation length was a reasonable assumption”?

P 18, caption extended data figure 2: “>” , “<” came out garbled in the pdf

p 14 Figure 3 What is the “amplitude of HPA”, is this α , or expectation value of HPA over all paths influencing grid cell, $\langle |HPA| \rangle$? In the legends for the lower two figures you write “ $\langle HPA \rangle$ ” but is is the change in HPA, i.e. ΔHPA rather than $\langle HPA \rangle$, or do I misunderstand this.

p 21, extended data figure 5. I guess the traces are vertically offset to visually separate them – can you add lines to show where the zero-anomaly line is for each station pair. Are the pairs arranged by inter-station azimuth. Maybe add the azimuth to the label (or otherwise visualise it)

Below are the point-by-point responses to the comments of the reviewer #1:

Thank you for taking the time to go thoroughly through all the reviewer comments and addressing most of them. Overall, the paper is in quite good shape now. In particular, there is now a description of how the uncertainty of measurements was estimated. While the interpretation remains a tad speculative, it is certainly a plausible model, which can explain the anisotropy change observations. However, I am still a bit mystified by the explanation in the supplementary material (and partially in the rebuttal letter) on the regionalisation of the static anisotropy and its derivative, and what is being solved for. I am almost certain that this lack of understanding is only pointing to the clarity of explanation rather than a deeper issue, and blame might also be on my side for not being able to follow your description.

I will try to express my difficulties based on an example. The way I understand it, HPA is Ψ_P in eq 1, 2, and “orientation of the 2 Ψ anisotropy” is Ψ_α (it would be helpful to the reader if this were explicitly stated; and then the symbols used preferentially at least in the supplementary material).

Assuming a model with uniform anisotropy with fast north direction, and four stations arranged in a square:

A B ^

|

D C

according to equation 2,

$\Psi_P=0$ for paths A-B, D-C, A-D and B-C

$\Psi_P=-\alpha$ for D-B

and $\Psi_P=+\alpha$ for A-C

MS: This is correct except for the value of Ψ_P for D-B and A-C:

$\Psi_P(D-B) = - \Psi_P(A-C)$

but is not equal to the same α from eq.3 which is related to the amplitude of the anisotropy.

You write (l 393-394) “For every grid point of 2.2 km x 2.2 km, the average of the horizontal polarization anomalies is computed ...”. However, the average of the values for Ψ_P over the different paths would be 0 for the example above but this is misleading with regard to the true anisotropy; the same averaging to zero or nearly zero would also occur for very dense azimuthal coverage. Taking an average of random samples of an oscillation is generally not very informative.

MS: You are right and we are sorry for this confusion. We actually compute anisotropy parameters for each horizontal polarization anomaly HPA (with its corresponding sign and not its absolute value). Then we average the inverted anisotropy parameters in each grid cell. Hence we obtain an average orientation of anisotropy in a grid cell for the different HPA and the different paths through this cell. [Note that Ψ_P is the polarization anomaly ($\Psi_P = \text{deviated polarization} - \text{azimuth of the pair of receivers}$), it is not measured in the geographical coordinate system from the North but measured relative to the azimuth of the pair of receivers (= isotropic polarization).]

Thank you for pointing this out, we corrected this info in the text.

Further you write in the rebuttal letter “The technique presented in the paper does not allow to invert the amplitude of anisotropy (α)” but with the observations of Ψ_P at different azimuths you would have sufficient information to recover both α and Ψ_α . In a uniform model this would simply be a matter of fitting the phase and amplitude of a sine-wave based on a few Ψ_P measurements at different azimuths. There is no way to meaningfully estimate Ψ_α without also estimating α except in very contrived circumstances (if all the azimuths you had were very close to zero-crossing I guess you could have very low resolution of α and good resolution of Ψ_α)

Regionalisation is fundamentally a way of spatial averaging. As even the sign of Ψ_P depends on the propagation azimuth, and by necessity the regionalisation mixes paths with different azimuths, I don't understand what the regionalisation would represent. I could speculate that maybe you use the absolute value of Ψ_P , i.e. $|\Psi_P|$ for the regionalisation. This would be borderline OK, as on average this value would be proportional to α , but still you would get a bias depending on the actual mix of contributing azimuths. Solving for regionalised α and Ψ_α should still give you superior results in this situation, too.

Similar reasoning can be applied to equation 3 on the changes. $\Delta \Psi_P$ can be positive or negative depending on the azimuth of propagation, and if measurements from a few azimuths are available it should be possible to deduce both $\Delta \alpha$ and $\Delta \Psi_\alpha$.

MS: We do invert both orientation (Ψ_α) and amplitude (α) of anisotropy, since we invert the two azimuthal terms (from eq. 3 and also 2). But the amplitude is less constrained than the phase since it depends on the damping. Ψ_α is much better constrained.

Further comments on the rebuttal letter:

p2

> ... The same data error is

> taken for all pairs of stations, equal to 0.035rad (2degrees) and the a priori error on σ_Ψ is 25degrees. This choice is somehow arbitrary but dictated by what was

> observed in other tectonic contexts. It affects the amplitude of the inverted Ψ (and the

> posterior model errors) but not their lateral variations. In other words, whatever σ_Ψ ,

> the regions of large or small Ψ_P are at the same locations but with a slightly larger or

> smaller amplitude.

Thank you, this is helpful. I suggest to report this information (including the values used for a priori error in the supplementary material.

MS: Done

p 6 of rebuttal letter

>> “We have provided a full code availability statement in the manuscript” - I did not see this

>> statement in the manuscript. Only reference to code is in the author contribution

statement,

>> where it is written which author contributed which code. But it is not clear whether and how

>> these codes are accessible to scientists outside the working group.

> We probably misunderstood the statements about the availability of the data and will clarify it

> in the revised MS. We provided the link to download the seismic data:

> <http://www.bosai.go.jp/e/activities/database/>

\thank you for providing the link on seismic data availability (very good!), but what about “code availability”.

MS: Codes are available upon request. This info was corrected in the manuscript.

P 9 of rebuttal letter

> We consider that the change of seismic

> anisotropy induced by the Tohoku-Oki earthquakes is related to changes in the alignment of

> cracks that in turn open the permeability of the hydrothermal system. The initial increase of

> temperature of the hot spring waters (of $\approx 3^{\circ}\text{C}$) associated with the HPA change is consistent

> with such a permeability increase and a first shallow degassing.

In addition to a possible coseismic permeability increase, another effect is shaking induced compaction of porous layers, which then reduces porosity, in turn increasing hydraulic pressure. This is the effect behind liquefaction, but in a milder form will also lead to upward surges of fluids.

MS: You are right: liquefaction and compaction can both exist when shaking a porous medium, but our data cannot quantify these effects and our simple model does not address this issue.

Further comments regarding manuscript:

Main text

p 6 273 Add degree sign to temperature, i.e. 2°C

MS: Already done

Suppl Material:

In the whole added section, note that units like “km” should not be italicised.

MS: Okay, corrections done

P 15 l 393 grid point → grid cell **MS: Done**

l 399 “an empirical correlation length” - add which value you actually used in km (I guess 5 km ?) **MS: Done**

p 16 l 415-417 As written, the sentence does not make much sense. Did you mean to say “that this choice of correlation length was a reasonable assumption”? **MS: Yes, corrections done**

P 18, caption extended data figure 2: “>” , “<” came out garbled in the pdf **MS: corrected**

p 14 Figure 3 What is the “amplitude of HPA”, is this α , or expectation value of

HPA over all paths influencing grid cell, $\langle |HPA| \rangle$? In the legends for the lower two figures you write " $\langle HPA \rangle$ " but is is the change in HPA, i.e. ΔHPA rather than $\langle HPA \rangle$, or do I misunderstand this.

MS: The amplitude of HPA is the expectation value of HPA for paths passing through a grid cell. α represents the amplitude of anisotropy. We changed $\langle HPA \rangle$ to $\Delta(HPA)$, sorry for the confusion.

p 21, extended data figure 5. I guess the traces are vertically offset to visually separate them – can you add lines to show where the zero-anomaly line is for each station pair. Are the pairs arranged by inter-station azimuth. Maybe add the azimuth to the label (or otherwise visualise it)

MS: Figure 5 was modified and the azimuth of each pair of receivers was added.